



# Development of the global hydro-economic model (ECHO-Global version 1.0) for assessing the performance of water management options

Taher Kahil[1], Safa Baccour[2], Julian Joseph[1], Reetik Sahu[1], Peter Burek[1], Jia Yi Ng[1,3], Samar Asad[1], Dor Fridman[1], Jose Albiac[1,4], Frank A. Ward[5], Yoshihide Wada[6]

[1] Water Security Research Group, International Institute for Applied Systems Analysis (IIASA), Laxenburg, Austria

[2] Department of Agricultural Economics, Finance and Accounting, University of Cordoba, Cordoba, Spain

[3] College of Environmental Sciences and Engineering, Peking University, Beijing, China

[4] Department of Economic Analysis, University of Zaragoza, Zaragoza, Spain

[5] Department of Agricultural Economics and Agricultural Business, Water Science and Management Program, New Mexico State University, United States

[6] Biological and Environmental Science and Engineering Division, King Abdullah University of Science and Technology, Thuwal, Saudi Arabia

*Correspondence to*: Taher Kahil (kahil@iiasa.ac.at)

**Abstract**

Water scarcity is one of the most critical global environmental challenges. Addressing this challenge requires implementing economically-profitable and environmentally-sustainable water management interventions across scales globally. This study presents the development of the global version of the ECHO hydro-economic model (ECHO-Global version 1.0), for assessing the economic and environmental performance of water management options. This global version covers 282 subbasins worldwide, includes a detailed representation of irrigated agriculture and its management, and incorporates economic benefit functions of water use in the agricultural, domestic and industrial sectors calibrated using the positive mathematical programming procedure alongside with the water supply cost. We used ECHO-Global to simulate the impact of alternative water management scenarios under future climate and socio-economic changes, with the aim of demonstrating its value for informing water management decision making. Results of these simulations are overall consistent with previous studies evaluating the global cost of water supply and adaptation to global changes. Moreover, these results show the changes in water use and water supply and their economic impacts in a spatially-explicit way across the world, and highlight the opportunities for reducing those impacts through improved water management. Overall, this study demonstrates the capacity of ECHO-Global to address emerging research and practical questions related to future economic and environmental impacts of global changes on water resources and to translate global water goals (e.g., SDG6) into national and local policies.



## 1 Introduction

Pressures on the availability of global freshwater resources have been mounting in the last decades due to the impacts of climate change (Rodell et al., 2018). At the same time, increasing water withdrawals from growing populations and economies globally have caused water scarcity in large areas of the world to increase in the recent past (Huang et al., 2021). Water scarcity is projected to further exacerbate in many regions of the world under future climate change and socio-economic development (Greve et al., 2018). Water scarcity could result in severe economic losses and environmental impacts such as groundwater depletion, water quality degradation, and biodiversity loss (Levintal et al., 2023). These impacts are often largest in areas with limited adaptive capacity to climate change and increase with the uncertainty of climate change projections (Dolan et al., 2021). Therefore, water scarcity has become one of the most critical environmental risks for human society, requiring the identification of appropriate water management options, that are not only technically feasible, but also consistent across spatial scales (local, national, global). This spatial consistency is particularly relevant to ensure environmental sustainability, economic efficiency, and social equity because the availability of water and related resources (land, energy, biodiversity) varies significantly at local scales, but global processes such as atmospheric moisture flows, trade dynamics, market adaptations, international water- and non-water-related treaties could result in global spillover effects (Haqiqi et al., 2023). The Global Commission on the Economics of Water (2024) suggests that the water cycle must be managed as a global common good in a collective way through concerted action in every country, transboundary collaboration, and for the benefits of all. However, the choice of global water management options has been so far informed mostly using hydrological models or simplified economic assessment models lacking a comprehensive representation either of the hydrological processes and technological constraints or the decision-making behaviors of water managers and users (Yoon et al., 2024).

Hydro-economic modeling (HEM) has evolved into a rigorous and flexible decision support tool for assessing the economic benefits of water across its alternative uses, and for identifying water management options to address the impacts of water scarcity. There have been, however, few global-scale HEM applications due to the focus of many hydro-economic models on water-related questions relevant or regulated at a local level and due to the computational burden models at larger spatial scales pose (Ortiz-Partida et al., 2023). The few available global scale hydro-economic models or analyses have explored key aspects of global water management, such as estimating the costs of adaptation measures required to ensure that all water demands are met (Ward et al., 2010), assessing the cost-effectiveness of some adaptation options to close the future water gap (Straatsma et al., 2020), analyzing the effects of irrigation water reallocation among several crops for improving groundwater





sustainability and economic efficiency in major groundwater-using countries (Bierkens et al., 2019), projecting future global urban water scarcity and potential supply expansion solutions (He et al., 2021), and exploring global transformation pathways for water, energy and land required under climate change impacts and mitigation scenarios and their cost implications (Awais et al., 2024). However, none of these studies has integrated the possibilities of allocating the multiple water sources (surface water, groundwater, nonconventional water) across sectors and scales or comprehensively represented the behavior of water

decision makers, including the choice of optimal combinations of water management options among a wide range of available options, the choice of irrigated crops and agricultural water management practices, the use and management of water for domestic and industrial purposes, the operation and planning of water infrastructure, and responses to policy instruments such as water prices, water quotas, and infrastructure subsidies, or the cost-benefit implications of those decisions.

To address some of the gaps described above, we developed a global version of the ECHO hydro-economic model (ECHO-

Global version 1.0). This extended and improved version of ECHO upgrades an earlier version, described in Kahil et al. (2018), by operating at the subbasin scale globally, including a more detailed representation of irrigated agriculture and its management, and accounting for both the benefits and costs of water use, enabling the assessment of the impact of globally-implemented water management options and the design of optimal combinations of those options. We used ECHO-Global to simulate the effect of alternative water management scenarios under future climate and socio-economic changes. The results

of these simulations enable assessing the global changes in water use and water supply and their economic impacts, comparing the adaptation responses of decision makers and the performance of water management options in different basins across the world, and identifying joint opportunities for reducing the global impact of water scarcity. The results shown in this paper aims mainly to highlight the benefits of ECHO-Global model development, but could also provide insights into where investments in the water sector should be prioritized and which additional national and local policy interventions are needed to achieve

global water-related goals (e.g., SDG6). It is important to note that despite its global coverage, ECHO-Global can be run for individual or several basins without the need to run it for all basins of the world. The rest of the paper is organized as follows. Section 2 presents the modeling framework, including an overview of the model structure and mathematical formulation, spatial delineation, and model database. Section 3 introduces the scenario analysis implemented to demonstrate the benefits of the model, and section 4 describes the results of scenario analysis. Finally, section 5 discusses the main findings and concludes

with possible future developments.



## 2 Modeling framework

### 2.1 Model structure

Figure 1: Schematic representation of the ECHO-Global model.





ECHO-Global is a bottom-up non-linear optimization model, which includes an economic objective function and a representation of the most relevant biophysical and technological constraints of the water system. The main modules of ECHO-Global are schematically shown in Figure 1. The objective function of ECHO-Global, as shown in the optimization module, is to maximize the net present value of the economic benefits of water-related economic activities (irrigation, households, industries) over a specified time horizon (e.g., a year, a decade, or more) across subbasins within river basins at the global

scale. In the economic module, the economic benefits from water use in the irrigation sector are determined by finding the optimal behavior of irrigated areas subject to a set of technical and resource constraints. The economic benefits from urban and industrial water uses are determined by measuring the social surplus derived from inverse water demand functions estimated using the Point Expansion approach (Griffin, 2016). Demand functions relate water use to the price of water and other explanatory variables such as income, climate, and household (Young and Loomis, 2014). The economic benefit

functions are calibrated using the positive mathematical programming (PMP) procedure to address the regional-scale aggregation and overspecialization problems (Baccour et al., 2022; Dagnino and Ward, 2012).

The subbasin units are created by intersecting river basin and country administrative boundaries (hereafter basin-country units or BCUs) and are linked within a reduced-form transboundary river network. This spatial delineation seeks to cover both the political boundaries of management policies and hydrological domains. The spatial delineation used in ECHO-Global, which

covers 282 BCUs across the world is shown in Figure 2 alongside the description of the procedure to delineate BCUs in section 2.3. Each BCU is treated as a single unit, meaning that water flows between spatial locations within a BCU are not considered (i.e., water availability is aggregated over a BCU). However, water can be transferred between BCUs pertaining to the same river basin, and each BCU can have inflow from upstream BCUs as well as discharge into downstream BCUs and/or a natural sink.

ECHO-Global includes basic representations of main biophysical and technological features of the water system at the BCU level, as shown in the hydrological and agricultural modules. These include representations of various water supply sources (surface water, groundwater, desalinated water and treated wastewater), sectoral water demands (irrigation, domestic and industrial), and infrastructure (surface water reservoirs, desalination plants, wastewater treatment plants, and water supply and irrigation systems). River basin hydrology is represented by a node-link network based on the principle of water mass balance

and flow continuity, defined in both flows and stocks. The flow variables tracked by the model are headwater inflow, streamflow, surface water diversion, groundwater pumping, water applied (i.e., withdrawn) and consumed, return flow to streams and aquifers, reservoir release, and reservoir evaporation. The stock variables tracked by the model are the reservoir





storage volumes. The GAMS optimization software is used for ECHO-Global development and scenario simulations (Brooke et al., 1988).

## 2.2 Mathematical formulation

An overview of the main equations in ECHO-Global is presented in the following sub-sections. In all equations, parameters are represented by lower case letters and variables are represented by capital letters.

### 2.2.1 Surface water balance

A reduced-form water mass-balance equation is used in ECHO-Global to balance supply and demand and ensure water conservation in each BCU and time-step. The flow continuity equation enables the hydrological connectivity within BCUs and between BCUs pertaining to the same river basin. The balances are defined for each flow node, $i$, and each stock node, $s$. The main flow variables, $X_i$, tracked by ECHO-Global are headwater inflow, streamflow, surface water diversion, groundwater pumping, non-conventional water use, water applied and consumed, return flows, reservoir release, and reservoir evaporation. The stock variables, $S_s$, tracked by ECHO-Global include reservoir storage volumes.

Total surface water inflows to each BCU are defined as the total annual flows at the headwater gauge. The inflows, $X_{h,t}$, at each headwater gauge, $h$ (a subset of $i$), in time $t$ are equal to the sum of local runoff $r_{h,t}$ and inflow from upstream BCUs $I_{h,t}$:

$$X_{h,t} = r_{h,t} + I_{h,t} \tag{1}$$

The streamflow in each BCU, $X_{v,t}$, at each river gauge, $v$ (a subset of $i$), in time $t$ is equal to the sum of flows over any upstream node $i$ whose activities impact that streamflow. These nodes include headwater inflow, river gauge, diversion, surface return flow, and reservoir release. The streamflow at each river gauge, which is required to be nonnegative, is defined as follows:

$$X_{v,t} = \sum_i b_{i,v} \cdot X_{i,t} \tag{2}$$

where $b_{i,v}$ is a vector of coefficients that links flow nodes $i$ to river gauge nodes $v$. The coefficients take on values of 0 for non-contributing nodes, +1 for nodes that add flow, and -1 for nodes that reduce flow.

The downstream discharge, $X_{d,t}$, at each downstream river gauge, $d$ (a subset of $v$), in each BCU and time-step must be greater than or equal to the minimum downstream flow requirements, $f_{i,t}$, needed to meet delivery obligations to downstream users and protect aquatic ecosystems as follows:

$$X_{d,t} \geq f_{d,t} \tag{3}$$





Water stock, $S_{res,t}$, at each reservoir, $res$ (a subset of $s$), in time $t$ is defined in the following equations:

$$S_{res,t} = S_{res,t-1} - \sum_L b_{L,res} \cdot X_{L,t} - \sum_e b_{e,res} \cdot X_{e,t} \tag{4}$$

$$S_{res,0} = bs_{res,0} \tag{5}$$

$$S_{res,t} \leq c_{res}^{max} \tag{6}$$

$$S_{res,t} \geq c_{res}^{min} \tag{7}$$

where equation (4) states that reservoir water stock in each BCU, $S_{res,t}$, is equal to its stock in the previous time period, $S_{res,t-1}$, minus both the net release (outflow minus inflow) from the reservoir, $X_{L,t}$, and reservoir evaporation, $X_{e,t}$. Evaporation depends on reservoir features and climatic factors. Both sets of parameters $b_{L,res}$ and $b_{e,res}$ are binary matrices linking reservoir stock nodes to reservoir release and evaporation nodes, respectively. Equation (5) defines initial reservoir water stock at $t = 0$, $bs_{res,0}$. Upper and lower bounds on reservoir water stock are defined in equation (6) and (7), respectively. Parameters $c_{res}^{max}$ and $c_{res}^{min}$ are reservoir maximum capacity and dead storage, respectively. Upper bound constraint guarantees that reservoir stock in each time period never exceeds its maximum capacity, while lower bound constraint states the capacity from which stored water in reservoir cannot be used.

### 2.2.2 Surface water diversion

Water supply to users in each BCU can be met partially or totally by diversions from a stream. However, during drought spells, streamflow can be low or even zero. Therefore, a surface water diversion constraint is required in order to avoid that diversion, $X_{d,t}$, exceeds available streamflow at each diversion node, $d$ (a subset of $i$), in time $t$. A diversion, which is required to be nonnegative, is defined as follows:

$$X_{d,t} \leq \sum_i b_{i,d} \cdot X_{i,t} \tag{8}$$

where $b_{i,v}$ is a vector of coefficients that links flow nodes, $i$, to diversion nodes, $d$. The right-hand side term represents the sum of all contributions to flow at diversion nodes from upstream sources. These sources include headwater inflow, river gauge, diversion, surface return flow, and reservoir release. The $b$ coefficients, take on values of 0 for non-contributing nodes, +1 for nodes that add flow, and -1 for nodes that reduce flow.





### 2.2.3 Groundwater pumping

Groundwater pumping originates from renewable and non-renewable sources. Renewable groundwater pumping, $X_{rp,t}$, is constrained by maximum monthly renewable (sustainable) supply, $gr_t$. Non-renewable groundwater pumping is physically unlimited, $X_{np,t}$, but it is considered a more expensive water supply source compared to surface water and renewable

groundwater. There is no modeled flow from groundwater to surface water. In future work, groundwater could be represented more comprehensively to better represent the effects of groundwater depletion based on e.g., the newly released global non-renewable groundwater withdrawals dataset of Niazi et al. (2024). However, our current approach allows evaluation of the sustainability of groundwater pumping given the projected use, and simulation of scenarios where maximum monthly renewable groundwater supply is adjusted to consider possible effects of groundwater depletion and climate change impacts.

Renewable groundwater pumping is defined in the following equation:

$$\sum_{rp} X_{rp,t} \leq gr_t \tag{9}$$

### 2.2.4 Non-conventional water use

The use of non-conventional water (desalinated water and treated wastewater), $X_{nc,t}$, is limited by the outflow from each non-conventional water supply technology as shown in equation (17) below. The use of desalinated water, $X_{d,t}$, is physically

unlimited in coastal areas. The use of treated wastewater, $X_{w,t}$, is limited by the available amount of urban and industrial water return flows, $X_t^{M\&I}$, as shown in the following equation:

$$\sum_{w} X_{w,t} \leq X_t^{M\&I} \tag{10}$$

### 2.2.5 Water applied, water consumption and return flows

Water applied, $X_{a,t}$, at each application node, $a$ (a subset of $i$), in time $t$ can stem from different supply sources $s$ (s subset of

$i$): surface water diversion, $X_{d,t}$, renewable groundwater pumping, $X_{rp,t}$, non-renewable groundwater pumping, $X_{np,t}$, and use of non-conventional water sources, $X_{nc,t}$. Water applied is defined as follows:

$$X_{a,t} = \sum_{s} b_{s,a} \cdot X_{s,t} \tag{11}$$

where $b_{s,a}$ is a vector of coefficients that link application nodes to supply source nodes. The coefficients take on values of 1 for application nodes withdrawing water from available sources, and 0 for not withdrawing water.

For each agricultural node in each BCU, total water applied for irrigation is defined as follows:



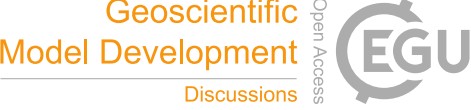

$$X_{a,t}^{ag} = \sum_{j,k} b_{a,j,k} \cdot \left( \sum_u b_{u,a} \cdot L_{u,j,k,t} \right) \tag{12}$$

Equation (12) states that irrigation water applied to crops from different water sources, $X_{a,t}^{ag}$, is equal to the sum over crops ($j$) and irrigation technologies ($k$) of water application per ha, $b_{a,j,k}$, multiplied by irrigated area, $L_{u,j,k,t}$, for each crop and irrigation technology. $L_{u,j,k,t}$ is multiplied by a binary matrix, $b_{u,a}$, to conform nodes.

Consumptive use, $X_{u,t}$, at each use node, $u$ (a subset of $i$), in time $t$ is an empirically determined proportion of water applied, $X_{a,t}$. For irrigation, consumptive use is the amount of water used through crop evapotranspiration (ET). For urban and industrial uses, consumptive use is the proportion of urban water supply not returned through the sewage system. That use, which cannot be negative, is defined as follows:

$$X_{u,t} = \sum_a b_{a,u} \cdot X_{a,t} \tag{13}$$

where parameters, $b_{a,u}$, are coefficients indicating the proportion of water applied that is consumptively used in each use node. For agricultural use nodes, water consumed is measured as:

$$X_{u,t}^{ag} = \sum_{j,k} b_{u,j,k} \cdot L_{u,j,k,t} \tag{14}$$

Equation (14) states that irrigation water consumed, $X_{u,t}^{ag}$, is equal to the sum over crops ($j$) and irrigation technologies ($k$) of empirically estimated ET per ha, $b_{u,j,k}$, multiplied by irrigated area, $L_{u,j,k,t}$, for each crop and irrigation technology.

Return flows, $X_{r,t}$, at each return flow node, $r$ (a subset of $i$), in time $t$ is a proportion of water applied, $X_{a,t}$. These flows return to the river system or contribute to aquifers recharge. Return flows are defined as follows:

$$X_{r,t} = \sum_a b_{r,a} \cdot X_{a,t} \tag{15}$$

where $b_{r,a}$ are coefficients indicating the proportion of total water applied that is returned to river and aquifers. For agricultural nodes, returns flows are defined as follows:

$$X_{r,t}^{ag} = \sum_{j,k} b_{r,j,k} \cdot \left( \sum_u b_{u,r} \cdot L_{u,j,k,t} \right) \tag{16}$$

Equation (16) states that irrigation return flows, $X_{r,t}^{ag}$, are equal to the sum over crops ($j$) and irrigation technologies ($k$) of empirically estimated return flows per ha, $b_{r,j,k}$, multiplied by irrigated area, $L_{u,j,k,t}$, for each crop and irrigation technology. $L_{u,j,k,t}$ is multiplied by a binary matrix, $b_{u,r}$, to conform nodes. The sum of water consumed and returned must be equal to water applied at each demand node.





### 2.2.6  Capacity of water supply technologies


A capacity constraint is used to limit the activity of the water supply sources $s$ according to the available physical capacity of the supply technologies $w$:

$$X_{s,t} \leq \sum_w b_{s,w} \cdot Z_{w,t} \tag{17}$$

where $Z_{w,t}$ is the installed capacity of each supply technology and $b_{s,w}$ are coefficients that link supply source nodes to supply

technology nodes. The capacity constraint therefore works, for instance, to ensure the volume of desalinated water produced does not exceed the installed desalination capacity or so that the volume of groundwater supplied via a pumping system does not exceed the installed capacity of that system.

Moreover, ECHO-Global incorporates capacity expansion decisions $Z_{w,t}^{new}$ that alleviate capacity constraints for the different water supply technologies including surface water reservoirs. Capacity retirements $Z_{w,t}^{ret}$ are further decision variables that

allow options to have finite lifecycles. The capacity expansion and retirement are currently considered exogenous decisions in ECHO-Global and can be adjusted through scenario simulations. The installed capacity of a particular option is thus given by:

$$Z_{w,t+1} = Z_{w,t} + Z_{w,t}^{new} - Z_{w,t}^{ret} \tag{18}$$

### 2.2.7  Economics

ECHO-Global also calculates the economic value of water for all uses of water based on the total willingness to pay of users

benefiting from them. For agricultural use, the economic value of water is measured by the contribution of water to farmers' net benefits. For urban and industrial uses, it is measured by the sum of the consumer and producer surplus.

Net benefits in each BCU, $NB_{u,t}$, at each use node $u$ in time $t$ is defined as follows:

$$NB_{u,t} = TB_{u,t} - TC_{u,t} \tag{19}$$

where $TB_{u,t}$ and $TC_{u,t}$ are the total benefits and costs at each use node $u$ in time $t$, respectively. Total costs include the

investment and operating cost of supplying water from surface water diversion, groundwater pumping and nonconventional water use.

For agricultural use nodes $ag$, total benefits, $TB_{ag,t}$, and total costs, $TC_{ag,t}$, in time $t$ are defined by the following equations:

$$TB_{ag,t} = \sum_{j,k} \left( p_{ag,j} \cdot Y_{ag,j,k,t}(L_{ag,j,k,t}) \right) \cdot L_{ag,j,k,t} \tag{20}$$





$$TC_{ag,t} = \sum_{j,k}(pc_{ag,j,k,t} + WC_{ag,j,k,t}) \cdot L_{ag,j,k,t} \tag{21}$$

where $p_{ag,j}$ is crop prices; $pc_{ag,j,k,t}$ is non-water production costs, $WC_{ag,j,k,t}$ is the water costs, and $L_{ag,j,k,t}$ is crop area.

$Y_{ag,j,k,t}$ is the yield of each crop $j$ equipped with irrigation technology $k$. Yield is specified as linear in the amount of land in production. The yield functions take the following form:

$$Y_{ag,j,k,t}(L_{ag,j,k,t}) = \alpha_{0,ag,j,k} + \alpha_{1,ag,j,k} \cdot L_{ag,j,k,t} \tag{22}$$

in which $\alpha_{0,ag,j,k}$ is the intercept of the function which depicts crop yield for the first unit of land brought into production, and

$\alpha_{1,ag,j,k}$ is the linear term of the function which depicts the marginal effect of additional land on average yield. These parameters of the crop yield function are calculated based on the first-order conditions of the agricultural profit maximization problem following the PMP procedure (Dagnino and Ward, 2012).

For urban and industrial use nodes, $M\&I$, total benefits, $TB_{M\&I,t}$, and total costs, $TC_{M\&I,t}$, in time $t$ are defined by the following equations:

$$TB_{M\&I,t} = \beta_{0,M\&I} + \beta_{1,M\&I} \cdot X_{M\&I,t} + \beta_{2,M\&I} \cdot X_{M\&I,t}^2 \tag{23}$$

$$TC_{M\&I,t} = \delta_{M\&I} \cdot X_{M\&I,t} \tag{24}$$

where equation (23) is the total benefits function with a quadratic specification (linear demand), with parameters $\beta_{0,M\&I}$, $\beta_{1,M\&I}$ and $\beta_{2,M\&I}$ for the constant, linear and quadratic terms, respectively. For urban and industrial use nodes, water is used first for high-valued uses such as indoor uses for drinking, sanitation, and cooking, so that benefits rise quickly for initial supplies

allocated to these uses. These high-value uses have few substitution possibilities, and therefore $\beta_{1,M\&I}$ is expected to be large and positive. However, urban and industrial marginal benefits fall rapidly for other additional low-value uses, such as outdoor uses for landscape irrigation, dust control, and car washing. Then $\beta_{2,M\&I}$ is expected to be large and negative. The water demand function is assumed to be linear and estimated based on Griffin (2016), with the extrapolation of the demand curve in the vicinity of an observed point where the price paid for water, the water quantity $X_{M\&I}$, and the price elasticity of demand

are known. Equation (24) represents total urban and industrial water supply costs, with $\delta_{M\&I}$ being the per unit cost of water supplied. It is important to note that estimating the economic benefits of water use in the industrial sector is not straightforward because of data limitations (e.g., lack of estimates of the marginal value of water), absence of market prices for water as water used within the sector is often self-supplied, and the difficulty to define the technical relationship between water use and output (Baker et al., 2021).





### 2.2.8    Objective function


To determine the optimal solution and the associated decision variables (optimized water flows and stocks, land use decisions and economic outcomes), ECHO-Global maximizes the net present value of the total net benefits of using water in all BCUs at the global scale over the planning horizon subject to the constraints (1) to (24). The length of the planning horizon depends upon the specific problem under consideration. The objective function of ECHO-Global takes the following form:

$$Max\ NPV = \sum_{u,t} \frac{NB_{u,t}}{(1+r)^t} \tag{25}$$

where $NPV$ is the net present value, $NB_{u,t}$ are the net benefits of each water use node $u$ in time $t$, and $r$ is the discount rate.

### 2.3  Spatial delineation and node-link network

Balancing spatial details with computational requirements is critical in ECHO-Global because the size of the optimization problem, as described in the previous section, can increase exponentially with the number of spatial units. Thus, to minimize

the computational burden, ECHO-Global runs at the level of BCUs representing the intersection between river basin and country administrative boundaries as shown in Figure 2. These BCUs are based on IFPRI's IMPACT-WATER model's "food-producing units" (Ledvina et al., 2013). These were created by dividing the globe into 106 river basins and then separately defining 116 economic regions (mainly countries), which identify the political boundaries of management policy. The selection and scale of these regions seeks to isolate the most important river basins and countries in terms of water use, especially for

irrigation purposes, and the 282 BCUs are then defined by their intersection. This procedure results in some international river basins being spread over several connected BCUs (e.g., the Indus is divided into 3 BCUs and the Nile is divided into 6 BCUs). On the other hand, many river basins are located within a single economic region (e.g., the Missouri Basin in the U.S.). This spatial delineation can be increased (e.g., increasing the number of BCUs in a river basins) in deep dive assessments without the need to significantly modify the core model mathematical formulation. The connections between BCUs pertaining to the

same river basin have been defined using a reduced-form river network, including a basic representation in each BCU of water supply (surface water, groundwater, non-conventional water) and water demand (agriculture, households, industries) nodes and major links between nodes (diversion, pumping, return flows). This network includes, for instance, 1410 river gauge nodes and 1128 demand nodes. Table A1 in the Appendices provides the list of river basins and countries included in ECHO-Global.





310         **Figure 2: ECHO-Global spatial delineation and schematic node-link network.**





## 2.4 Model database

Table 1 provides an overview of the data sources to parameterize ECHO-Global and their spatial and temporal resolutions.

**Table 1: Data sources for parameterization of the global version of ECHO-Global.**

| Parameters | Description | Data source | Spatial resolution | Temporal resolution |
|---|---|---|---|---|
| Water availability | Runoff, river discharge, and groundwater recharge, environmental flow (average from 4 climate models, GFDL-ESM2M, HadGEM2-ES, IPSL-CM5A-LR, MIROC5) | CWatM model simulations (Burek et al., 2020) | 0.5° × 0.5° | Daily for 2010 (average 2006-2015) -2050 (average 2046-2055) |
| Water demand | Monthly domestic and industrial water demands | WFaS dataset (Wada et al., 2016) | 0.5° × 0.5° | Daily for 2010 (average 2006-2015) -2050 (average 2046-2055) |
| | Recycling ratios for domestic and industrial water | Wada et al. (2014) | National | Yearly for 2010-2050 |
| | Crop-specific calendars | MIRCA2000 (Portmann et al., 2010) | 5′ × 5′ | Daily for 2000 |
| | Potential evapotranspiration, effective precipitation | CWatM model simulations (Burek et al., 2020) | 0.5° × 0.5° | Daily for 2010 (average 2006-2015) -2050 (average 2046-2055) |
| | Irrigation efficiency | FAO-AQUASTAT database | National | 2010 |
| Water infrastructure | Reservoir capacity | Global Reservoir and Dam Database (GRanD) (Lehner et al., 2011) | Asset level | 2011 |
| | Reservoir area-capacity function slope | Yigzaw et al. (2018) | Asset level | 2011 |
| | Coefficient of reservoir evaporation loss | CWatM model simulations (Burek et al., 2020) | 0.5° × 0.5° | Daily for 2010 (average 2006-2015) -2050 (average 2046-2055) |
| | Surface water diversion and groundwater pumping capacity | PCR-GLOBWB model simulations (Wada and Bierkens, 2014) | 0.5° × 0.5° | Daily for 2010 (average 2006-2015) |
| | Desalination capacity | DESALDATA (Global Water Intelligence, 2017) | Asset level | 2010 |
| | Wastewater treatment capacity | Jones et al. (2021) | National | 2015 |





| Economic data | Crop prices | FAO-FAOSTAT database | National | 2010 (average 2006-2015) |
|---|---|---|---|---|
| | Crop areas, Crop yields | MAPSPAM (Yu et al., 2020) | 5′ × 5′ | 2010 |
| | Crop non-water production costs | Sauer et al. (2010), U.S. Department of Agriculture (USDA ERS - Commodity Costs and Returns, 2024), Vittis et al. (2021) | Different resolutions (national, global, etc.) | 2010 (average 2006-2015) |
| | Water prices for domestic and industrial water uses | International Benchmarking Network for Water and Sanitation Utilities (IBNET) database | National | Latest available data |
| | Elasticity of demand for domestic and industrial water use | Reynaud and Romano (2018), Gracia-de-Rentería and Barberán (2021) | Different countries | Latest available data |
| | Investment and O&M cost of water supply from different water sources | Kahil et al., (2018) | Different resolutions (national, global, etc.) | Latest available data |

### 2.4.1 Estimation of water availability and demand

The total average monthly data at BCU level for current (time period 2006-2015) and future (time period 2046-2055) conditions of several water availability parameters including runoff, discharge and groundwater recharge are estimated to act as nodal inputs into the node-link network of ECHO-Global, based on simulations conducted by the hydrological model CWatM (Burek et al., 2020), that provides a grid-based representation of terrestrial hydrology, applied globally at a spatial resolution of 30 arcmin (~50 km) and daily temporal resolution using climate forcing data from 4 different climate models.

Environmental flow requirements in each BCU are estimated using CWatM simulations based on the Pastor et al. (2014) Variable Monthly Flow (VMF) method. To aggregate the grid-based results of CWatM into the BCU spatial delineation of ECHO-Global, the BCU polygons are rasterized in a preprocessing step on a 30-arcmin grid, to compute the water availability in all grid cells within the BCU and in all grid cells that are upstream of those grid cells. Figure 3 shows the change in runoff between current and future conditions at the BCU level based on CWatM simulations.

Monthly sectoral water demands at BCU level for current (time period 2006-2015) and future (time period 2046-2055) conditions are estimated to be included as inputs into ECHO-Global. Monthly irrigation water demands are estimated for each BCU using irrigated crop area and monthly gross water requirements per unit area. In order to estimate irrigated crop area in each BCU, data on harvested area (year 2010) for 13 irrigated crops at the global scale with a spatial resolution of 10 km are obtained from the MAPSPAM dataset (Yu et al., 2020). This gridded crop area is aggregated across each BCU. Net water

requirements for irrigation per unit crop area (i.e., consumptive demands) are estimated using the crop coefficient method



(Allen et al., 1998). Monthly crop evapotranspiration is calculated by combining a crop coefficient per crop development stage with a monthly reference (potential) evapotranspiration. Net monthly irrigation requirements are calculated at BCU level, so as to ensure the optimum growth of each crop. These net requirements are the difference between crop evapotranspiration and effective precipitation. Crop-specific calendars and crop coefficients are obtained from the MIRCA2000 dataset (Portmann et

al., 2010), while current and future potential evapotranspiration and effective precipitation are taken from CWatM simulations. Lastly, irrigation water gross requirements are calculated per unit crop area and at BCU level as the ratio between irrigation water net requirements and irrigation efficiency. This efficiency factor measures the overall effectiveness of irrigation, which takes into account losses during water conveyance as well as application efficiency at plot level. Current levels of irrigation efficiency are obtained from FAO-AQUASTAT database. Irrigation return flows are computed as the difference between gross

and net irrigation requirements. Monthly domestic and industrial water demands are calculated using the Water Futures and Solutions (WFaS) dataset (Wada et al., 2016) that provides global projections of water demand at a spatial resolution of 50 km and daily temporal resolution for current and future conditions under various climate and socio-economic scenarios. The volume of return flows from both the domestic and industrial sectors is determined by recycling ratios developed per country taken from Wada et al. (2014).


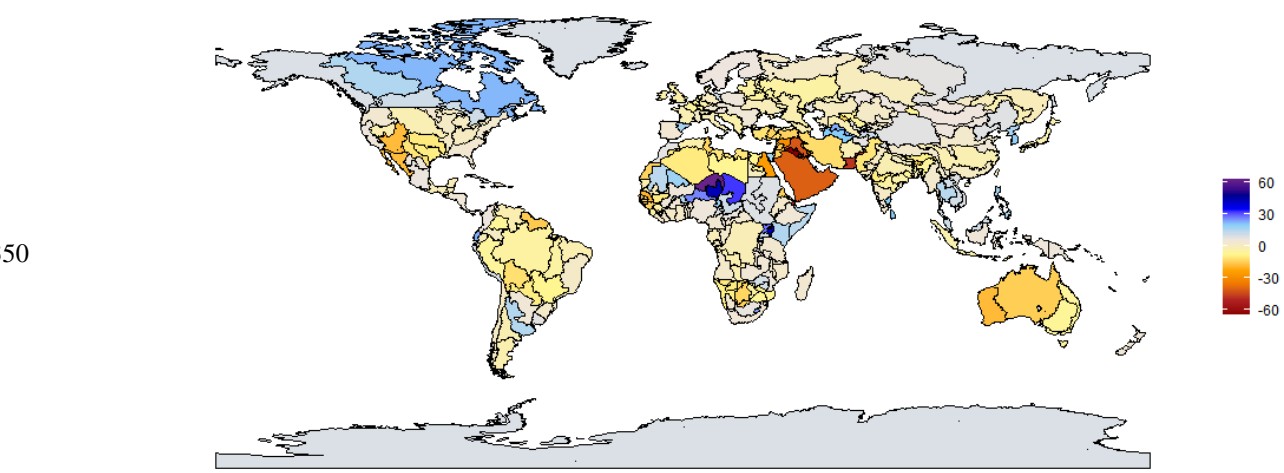


**Figure 3: Runoff change between 2010 and 2050 based on CWatM simulations using average climate forcing data from 4 GCMs under RCP6.0**





### 2.4.2    Existing capacity of water management infrastructure

The existing capacity of the different water infrastructure (e.g., reservoirs, surface water diversion, groundwater pumping,
wastewater treatment and desalination plants) implemented in ECHO-Global is assessed at the BCU level based on information
gathered from various databases. The capacities of existing surface water reservoirs are estimated by aggregating facility-level
data from the GRanD database (Lehner et al., 2011). Evaporative losses due to increased surface area during reservoir storage
are incorporated into the water mass-balance equation defined in section 2.2 using a linearized storage-area-depth relationship
developed based on the dataset of Yigzaw et al. (2018). The existing capacities of surface water diversion and groundwater
pumping infrastructure are identified using historical gridded water withdrawals and groundwater extraction rates from Wada
and Bierkens (2014). These withdrawals are aggregated to the level of the BCUs, and the maximum monthly withdrawal in
the historical time-series plus a 10% reserve margin is used to define the capacity in each BCU. Existing desalination capacities
are identified using a refined version of the global desalination database (DESALDATA) (GWI, 2017). Wastewater treatment
and reuse capacities are defined using estimates of return flows from the domestic and industrial sectors and country level data
and wastewater production, collection, treatment and reuse from Jones et al. (2021). The existing water treatment capacity is
estimated in each BCU by multiplying the estimated rates of water treatment (i.e., wastewater treated/wastewater produced)
and reuse (i.e., wastewater reuse/wastewater produced) for 2015 by the maximum volume of domestic and industrial return
flows calculated in ECHO-Global.

### 2.4.3    Economic data

A significant amount of economic data associated with the economic activities and water management options considered are
required to parametrize ECHO-Global. For irrigated agriculture, country-specific prices of 13 crops, representing 89% of
global irrigated area, are retrieved from the FAOSTAT database, while crop areas and crop yields are obtained from the
MAPSPAM dataset (Yu et al., 2020). Non-water production costs of those crops are estimated based on several studies in the
literature. For domestic and industrial activities, we use downward sloping demand functions of water price with constant
elasticity, to model consumer and producer surpluses. The self-price elasticities of domestic (assumed to be -0.1) and industrial
(assumed to be -0.54) water uses are taken from the literature, although elasticity estimates can be highly variable, depending
on economic, political and environmental conditions. Observed water prices for domestic and industrial water uses are taken
from the International Benchmarking Network for Water and Sanitation Utilities (IBNET) database. However, those water
prices are often set below market clearing prices, which results in a misestimation of the demand function. Information on the
investment and operating cost of different water supply sources and technologies (surface water diversion, groundwater





pumping, reuse of treated wastewater, desalinated water, surface water reservoirs, irrigation systems) are taken from Kahil et al. (2018) based on an extensive literature review.

## 3    Water management scenarios

A set of global water management scenarios have been developed for the year 2050 based on changes in several driving factors that encompass both climatic and socioeconomic conditions and choices of water management strategies as shown in

Table 2. The projected changes on water supply and demand for 2050 are based on the global water scenarios that combine the Shared Socio-economic Pathways (SSPs) and Representative Concentration Pathways (RCPs) developed by Wada et al. (2016). In this paper, we explore strategies that enhance water resources management under the SSP2-RCP6.0 scenario. The different water management scenarios aim to demonstrate to what extent water demand and supply management strategies can mitigate future climate and socio-economic change impacts and highlight the ability of ECHO-Global to assess the economic and environmental impacts of adaptation strategies. Five alternative scenarios for 2050 under the SSP2-RCP6.0 are assessed in our study, each representing different management options, ranging from a business-as-usual (BAU) to a more sustainable scenario (RES). The BAU scenario includes the future projections of water availability and demand for 2050, and reflects the continuation of current water use and management practices. The environmental sustainability (ENV) scenario integrates environmental flow requirements and minimizes the use of non-renewable groundwater. The preservation of environmental flow acknowledges the importance of maintaining adequate water flow for ecological health alongside water usage. The demand management (DM) scenario identifies an optimal allocation of water and land to enhance agricultural water use efficiency. The supply management strategies are incorporated into two scenarios: the expansion of non-conventional water use (NC) and of reservoir storage capacity (RES). The NC scenario entails incorporating additional non-conventional water supply capacity, namely wastewater recycling and desalination, alongside surface- and ground-water sources to fulfill future water demand. The RES scenario simulates the effect of increasing reservoir storage capacities.

**Table 2: Summary of the water management scenarios.**

| Scenarios | |
|---|---|
| **Water availability** | Runoff, groundwater recharge, and evaporation for 2050 is projected using the hydrological model CWatM based on average climate forcing data from 4 GCMs under the climate change scenario RCP6.0. |
| **Water demand** | Water demand of agricultural, urban, and industrial sectors are projected for 2050 based on assumptions about GDP growth, population growth, technological development, and change in climatic parameters for the SSP2-RCP6.0 scenario. |
| **Policy constraints for the water management scenarios** | |





| | BAU<br>*Business as usual* | ENV<br>*Environmental sustainability* | DM<br>*Demand management* | NC<br>*Non-conventional sources* | RES<br>*Increased reservoir storage capacity* |
|---|---|---|---|---|---|
| **Groundwater Use** | No limit on non-renewable groundwater use. | Minimizing non-renewable groundwater use. | Constraint limiting use of non-renewable groundwater. | Constraint limiting use of non-renewable groundwater. | Constraint limiting use of non-renewable groundwater. |
| **Environmental flow** | No constraint. | Environmental flow constraint. | Environmental flow constraint. | Environmental flow constraint. | Environmental flow constraint. |
| **Crop allocation** | Proportional allocation (i.e., equal relative change) of crop land area. | Proportional allocation of crop land area. | Optimal allocation of crop land area driven by crop economic value. | Optimal allocation of crop land area driven by crop economic value. | Optimal allocation of crop land area driven by crop economic value. |
| **Sectoral water allocation** | Constraint prioritizing water use for urban and industrial sectors over agriculture. | Constraint prioritizing water use for urban and industrial sectors over agriculture. | Optimal water allocation among sectors driven by the economic value of water in each use. | Optimal water allocation among sectors driven by the economic value of water in each use. | Optimal water allocation among sectors driven by the economic value of water in each use. |
| **Desalination** | Constraint limiting use of desalination to current capacity in coastal basins. | Constraint limiting use of desalination to current capacity in coastal basins. | Constraint limiting use of desalination to current capacity in coastal basins. | No limit on desalinated water use in coastal basins. | No limit on desalinated water use in coastal basins. |
| **Use of treated wastewater** | Constraint limiting use of wastewater to current capacity. | Constraint limiting use of wastewater to current capacity. | Constraint limiting use of wastewater to current capacity. | Increased wastewater capacity based on wastewater produced under DM scenario. | Increased wastewater capacity based on wastewater produced under DM scenario. |
| **Irrigation efficiency** | No improvement in current levels of irrigation efficiency | No improvement in current levels of irrigation efficiency | Increase irrigation efficiency in BCUs to maximum efficiency level for each basin. | Increase irrigation efficiency in BCUs to maximum efficiency level for each basin. | Increase irrigation efficiency in BCUs to maximum efficiency level for each basin. |
| **Reservoir storage capacity** | Constraint limiting reservoir storage capacity to current capacity. | Constraint limiting reservoir storage capacity to current capacity. | Constraint limiting reservoir storage capacity to current capacity. | Constraint limiting reservoir storage capacity to current capacity. | Increase reservoir storage capacity by 50% in BCUs suffering from water deficits limited by maximum storage potential based on Liu et al. (2018). |





## 4   Results

### 4.1  Model validation

To ensure the accuracy and reliability of the ECHO-Global model and its capacity to produce robust future projections, simulated water use by sector and source, irrigated area and agricultural income at country level have been calibrated and validated, for the base year 2010. The calibration process consists in adjusting model parameters such as irrigation efficiency, gross crop water requirements and water supply costs, and using upper and lower bound constraints for some model variables such as urban and industrial water withdrawals or non-conventional water use. The domestic and industrial water withdrawals are validated using the WFaS dataset, while irrigation water use is based on reported values in the FAO-AQUASTAT database. The irrigated agriculture income is validated using MAPSPAM dataset. Figure 4 displays the observed and simulated global water use by sector and source, irrigated area and agricultural income by crop type, and the 10 countries with the highest values in 2010. Overall, the results in Figure 4 indicate the ECHO-Global results in terms of water use, irrigated crop area and irrigated agriculture income deviate by 2-13% from the observed values, indicating an acceptable level of reliability and thus suitability to be used for simulation of alternative scenarios and policy interventions.

The simulated global water withdrawals amount to 3,741 km$^3$/year, 2% less than the observed value. In 2010, the largest water withdrawals are found in India, China, the United States, and Pakistan, exhibiting a 3-7% difference compared to the observed withdrawals. The simulated water withdrawals for the domestic and industrial sectors are 1% lower than the observed data and estimated at 425 and 835 km$^3$/year, respectively. The model accurately estimates irrigation water withdrawals at 2,480 km$^3$/year, 3% less than the observed data. The simulated surface and non-conventional water withdrawals closely align with the observed values in 2010 and are estimated at 2,980 and 39 km$^3$/year in 2010, respectively. However, the simulated groundwater withdrawals are 17% less than the observed data and amount to 722 km$^3$/year.

The simulated global irrigated area amounts by 233 million ha in 2010, which is 6% lower than the observed value. The most important irrigated areas are in India, China, the United States, and Pakistan, which are 6-11% lower than the observed irrigated area. The main irrigated crop areas are rice (90 million ha), wheat (55 million ha), maize (27 million ha), and vegetables (16 million ha), and they are 1-9% lower than the observed values. The total agricultural income amounts to 435 billion USD/year, 13% lower than the observed values from the MAPSPAM dataset. Most of this income is generated from agricultural activities in the countries with the highest irrigated areas such as India, China, and the United States.





**Figure 4: (a) The simulated and observed total water withdrawals by sector and water source at the global scale in 2010 and the ten countries with highest withdrawals. (b) The simulated and observed irrigated area at the global scale in 2010 and the ten countries with highest irrigated areas. (c) The simulated and observed agricultural income at the global scale in 2010 and the ten countries**





**with the highest agricultural incomes. Full names for countries are provided in Table A1 in the appendices. Crop full names are: Wht= Wheat, Ric= Rice, Mai= Maize, Ocr= Other cereals, Cot= Cotton, Ofb= Other fibers, Rot= Root crops, OilC= Oil crops, Frt= Fruit trees, Veg= Vegetables.**

## 4.2 Scenarios simulation

### 4.2.1    Water withdrawals

Figure 5 shows water withdrawals by sector (agriculture, domestic and industrial) and sources of water (surface water, groundwater, treated wastewater and desalination) and the 10 countries and basins with the highest changes in withdrawals between 2010 and 2050 across the different water management scenarios. Figure 6 depicts water withdrawals at the BCU level

in 2010 and shows the impacts of management scenarios on water withdrawals in 2050. Results indicate that global water withdrawals amount to 3,730 Km$^3$/year in 2010 and are expected to rise by 30% to 4,860 Km$^3$/year by 2050 under the BAU scenario. As expected, the alternative water management scenarios (ENV, DM, NC, RES) result in a decrease of total water withdrawals to 3,560-4,280 Km$^3$/year by 2050, a reduction of 12-27% compared to the BAU scenario. The DM scenario shows the highest reduction in water withdrawals due to improved irrigation efficiency and optimized irrigated crop area, as well as

optimal domestic and industrial water use. Results also show large spatial heterogeneity in water withdrawals globally, with the most considerable increases in water withdrawals between 2010 and 2050 in all scenarios are expected to occur in China, India, and Russia by country and in the Chang Jiang, Ganges, Huang He, and Hual He by basin, because of increased domestic and industrial water demands and irrigation water requirements. On the other hand, the most substantial decreases in water withdrawals between 2010 and 2050 in all scenarios are expected to occur in the United States, Germany, Uruguay, and Japan

by country and in the Rhine, Mississippi, Great Lakes, and Uruguay by basin. This is mainly due to a reduction in industrial water demand in most locations, as well as decreased water availability in some countries such as Japan, Pakistan, India, and Iran. In 2050, the ENV, DM, NC, and RES scenarios demonstrate considerable opportunities for conserving water resources, particularly in China, India, the United States, Pakistan, and Russia, when compared to the BAU scenario.

The global industrial and domestic withdrawals are projected to rise considerably in all scenarios from 1,250 km$^3$/year in 2010

to 2,110-2,340 km$^3$/year in 2050. Most increases in these withdrawals are expected to take place in China, India, Russia, and Indonesia by country, and in Chang Jiang, Huang He, Ganges, and Zhu Jiang by basin. Irrigation withdrawals are expected to grow slightly from 2,480 km$^3$/year in 2010 to 2,520 km$^3$/year in 2050 under the BAU scenario, because of climate change impact, without considering the potential for expanding irrigated areas. Major increases in irrigation withdrawals under the BAU in 2050 are found in India, the United States, Pakistan, Iran, and China by country, and in Indus, Krishna, Ganges, and

Huang He by basin. Irrigation water withdrawals are expected to fall under the ENV, DM, NC, and RES scenarios to 1,445-1,940 km$^3$/year in 2050. The most gains from improved water management in 2050 are projected to take place in India, China,





Pakistan, Iran, and Scandinavia by country, and in the Ganges, Indus, Western Asia Iran, and Scandinavia by basin. Irrigation will continue to be the largest global water user under all scenarios, but its relative share is expected to decline to 41-52% by 2050.

Several water sources are used to fulfill the water withdrawals for all sectors. Surface water is the main source of water used in all scenarios. Surface water remained at the current level of 2,980 km$^3$/year for the BAU scenario, while it decreased by only 1 km$^3$/year for the ENV scenario, to around 2,800 km$^3$/year for the DM and NC scenarios, and to 2,880 km$^3$/year for RES scenario. This decrease in surface water withdrawals can be attributed to improved irrigation efficiency and better water allocation within and among sectors and BCUs. Major decreases in surface water withdrawals are found in Scandinavia, India,

Japan, and Philippines by country, and in Scandinavia, Japan, Thai Myan Malay, and Philippines by basin. Groundwater pumping rises by 160% from 709 km$^3$/year in 2010 to 1,844 km$^3$/year in 2050 under the BAU scenario. The increase in groundwater depletion can be attributed to the growing water demand and the lack of constraints on non-renewable groundwater use. This trend is primarily noticeable in China, India, Russia, and Nigeria. Among the various scenarios aimed at minimizing non-renewable groundwater use, the ENV scenario achieved a reduction of 30% to 1,260 km$^3$/year of

groundwater pumping, and DM, NC, and RES scenarios decreased groundwater use by around 60%, corresponding to a range of 729-690 km$^3$/year, compared to BAU scenario in 2050. These decreases in groundwater use are projected to take place in India, the United State, Pakistan, Iran, and Gulf by country, and in Indus, Ganges, Western Asia Ira, Arabian Peninsula, and California by basin. The use of non-conventional water (desalination and treated wastewater) amounts to about 38 km$^3$/year in 2010 and increases by only 1 km$^3$/year for the ENV and DM scenarios in 2050. An expansion of desalination and wastewater

treatment and reuse capacities under the NC and RES scenarios help in fulfilling the demand growth, eventually leading to an increase of non-conventional water use to 94 km$^3$/year by 2050. Desalination use is expected to surge in coastal areas of Israel, Egypt, and Bangladesh, while treated wastewater use is expected to expand in China, India, Niger, Iran, and the Gulf countries.

Results suggest that demand management options are critical for the conservation of water resources, efficient allocation of water among and within sectors and BCUs, reduction of the environmental impact of growing water use, and ensuring a reliable

water supply for future generations. Additionally, these options can help in adapting to the impacts of climate change. The rate of adoption of the different demand management options varies among BCUs and scenarios, but a higher adoption rate is necessarily found in areas that are facing challenges related to reduced water availability and growing water demand.

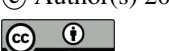



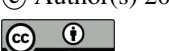

**Figure 5: (a)** Global water withdrawals by water use sector and water source for each scenario. **(b)** Ten countries (left column) and basins (right column) with highest change in withdrawals between 2010 and 2050 for each scenario. Non-conventional water includes both desalinated water and treated wastewater. Full names for countries and basins are provided in Table A1 in the appendices.





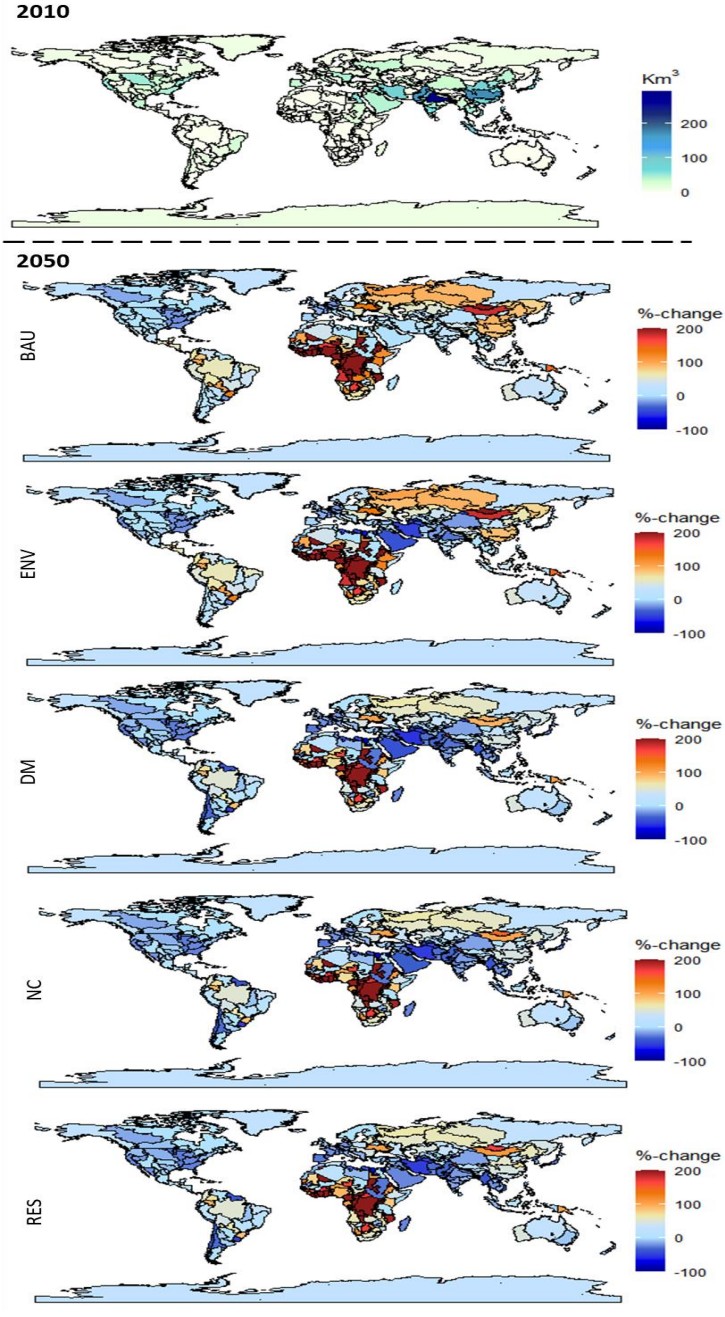

**Figure 6: Water withdrawals at the BCU level in 2010 and percentage change of withdrawals in 2050 compared with 2010 for each scenario. The list of basins and countries is provided in Table A1 in appendices.**



### 4.2.2 Irrigated area

Figure 7 depicts irrigated area by crop in 2010 and in 2050 for each scenario and the 10 countries and basins with the highest changes in total irrigated area across the different water management scenarios. The total irrigated area amounts to 233 million

ha in 2010 and is projected to decrease in all scenarios by 2050 due to the impact of climate change on water availability and crop water requirements and growing competition with domestic and industrial water uses. The BAU scenario slightly reduces irrigated area by 2% to 229 million ha, while the enforcement of environmental flows under the ENV scenario substantially reduces irrigated area by 27% to 170 million ha in 2050. The reduction in irrigated area is projected to occur in China, India, Pakistan, and Iran by country, and in the Ganges, Huang He, Indus, and Western Asia Iran by basin for the ENV scenario. The

enhancement of environmental flows alongside the implementation of demand and supply management options (DM, NC, RES) would have a lower reduction in irrigated areas compared to the ENV scenario. The demand management options (in the DM scenario) reduces irrigated areas by 14% to 199 million ha while the supply enhancement options (NC, RES) only decreased irrigated areas by 9% to 212 million ha, compared to the 2010 irrigated area. The potential of demand management and supply enhancement options (DM, NC, RES) to address the reduction of irrigated areas is predominantly observed in

China, India, Iran, and Egypt.

Results show that the decrease in irrigated areas across all scenarios mainly affects crops such as wheat, maize, and other cereals, which are the major crops globally, but often have lower market values. To minimize the impact on low value crops, a proportional reduction in irrigated crop area is implemented for each BCUs under the BAU and ENV scenarios. Results indicate that the ENV scenario would reduce cereals (wheat, maize, and other cereals) by 36-45%, cotton by 32%, oil crops

by 27%, roots by 26%, fruit by 24%, vegetables by 21%, and rice by 14% in 2050. This approach strikes a balance between efficient water allocation, reduced risks from crop overspecialization, and food security requirements, recognizing the varying economic importance of different crops within the agricultural system. An optimal allocation of irrigated areas is implemented under the DM, NC, and RES scenarios to maximize the economic efficiency of water use. As expected, the optimal allocation of crop land leads to relatively lower reductions for crops such as cotton, roots, fruits, and vegetables compared to the

proportional land reduction. These crops generally have high market values and low water requirements. The DM, NC, and RES scenarios reduce the area of cereals by 16-40%, cotton by 10-13%, oil crops by 12-20%, roots by 5-7%, fruit by 5-6%, vegetables by 2-3%, and rice by 2-3% in 2050.







**Figure 7: (a) Total irrigated area, actual irrigated cropland distribution, and irrigated cropland at the global scale for each scenario.**
**(b) Ten countries (left column) and basins (right column) with highest change in total irrigated area between 2010 and 2050 at the global for each scenario. Full names for countries and basins are provided in Table A1 in the appendices. Crop full names are: Wht= Wheat, Ric= Rice, Mai= Maize, Ocr= Other cereals, Cot= Cotton, Ofb= Other fibers, Rot= Root crops, OilC= Oil crops, Frt= Fruit trees, Veg= Vegetables.**



### 4.2.3    The costs and benefits of water use

Figure 8 depicts the annual gross benefits of water use in the agricultural, industrial, and domestic sectors in 2010 and 2050 for each scenario, the total operational and investment costs of water supply sources, irrigation efficiency, and reservoir expansion, and the 10 countries and basins with the highest changes in gross benefits and water costs across the different water management scenarios. Results show that the global gross benefits across all sectors and spatial locations amount to 4,378 billion USD/year, with a water cost of about 323 billion USD/year, resulting in a net benefit of 4,055 billion USD/year in 2010. The total gross benefits rise considerably to 6,571 billion USD/year in 2050 under the BAU scenario, driven by the growth in the domestic sector (65%), followed by the industrial sector (32%), and irrigated activities (3%). Despite an annual increase in water costs of 443 billion USD to reach 766 billion USD/year, the BAU scenario yields additional net benefits of 1,750 billion USD per year compared to 2010. The ENV scenario delivers additional annual net benefits of 1,726 billion USD compared to 2010, which are slightly less than those in the BAU (-1.4%). The annual net benefits for the DM, NC, and RES scenarios rise by approximately 1,760, 1,761, and 1,766 billion USD, respectively, compared to 2010, fully offsetting the cost of the environmental constraints implemented in the ENV scenario compared to the BAU. The increase in gross and net benefits is projected to take place in China, India, Scandinavia, and Central Europe by country, and in Ganges, Chang Jiang, Scandinavia, and Huang He by basin. The total gross benefits are projected to fall slightly under the ENV, DM, NC, and RES scenarios to 6,443-6,546 billion USD/year in 2050, a decrease of 0.4-2% compared to the BAU scenario. However, the total net benefits increase for the demand and supply management scenarios. In 2050, the DM scenario increases net benefits by 10 billion USD/year, the NC scenario by 11 billion USD/year, and the RES scenario by 16 billion USD/year compared to the BAU scenario.

The total water costs in the baseline scenario amount to 323 billion USD/year, most of it for supplying surface water. The total water costs increase by 137% to around 766 billion USD/year under the BAU and ENV scenarios by 2050. This considerable rise is due to increased industrial and domestic water demand and crops water requirements, leading to a substantial rise in groundwater pumping costs. The DM scenario increases water use efficiency, reducing the water costs to 628 billion USD/year in 2050, while the additional use of desalination and treated wastewater under the NC scenario slightly increases the water costs to 632 billion USD/year in 2050. Expanding reservoir capacity under the RES scenario increases the water costs to 642 billion USD/year in 2050. The increase in water costs is projected to take place in China, India, and Russia by country, and in Chang Jiang, Huang He, and Zhu Jiang by basin.

The domestic sector generates 55% of the total gross benefits in 2010 (2,390 billion USD/year), followed by the industrial and agriculture sector, which contribute around 41% (1,800 billion USD/year) and 4% (190 billion USD/year), respectively. In



terms of water costs, the industrial sector has the highest share, representing 63% (203 billion USD/year) of the total costs. The domestic sector accounts for 30% (97 billion USD/year) of the water costs, while the agricultural sector has a smaller share of 7% (23 billion USD/year). In 2050, the net benefits from the domestic and industrial sectors are projected to increase by 72% and 6%, respectively, while it decreases slightly (-5%) for agriculture under the BAU scenario. The enforcement of environmental flows under the ENV scenario mainly affects irrigation activities, reducing the net benefits by 12% to 164

billion USD/year in 2050. However, the management options implemented in the DM, NC, and RES scenarios increase agricultural net benefits by 1-3% compared to the BAU scenario. The DM and NC scenarios boost the domestic sector's net benefits by 32 and 36 billion USD/year, and the agriculture net benefits by 5 and 2 billion USD/year, respectively while it reduces the industrial net benefits by 6 and 5 billion USD/year in 2050, respectively, compared to the BAU scenario. The RES scenario increases the domestic, industrial and agriculture's net benefits by 46, 7, and 2 billion USD/year, respectively, in 2050

compared to the BAU scenario.





**Figure 8: (a) Annual economic gross benefits of water use in the agricultural, industrial, and domestic sectors. (b) Total costs of water technologies, irrigation efficiency, and reservoir expansion. (c) Ten countries (left column) and basins (right column) with highest change in annual gross benefits between 2010 and 2050 at the global scale for each scenario. (d) Ten countries (left column) and basins (right column) with highest change in annual water costs between 2010 and 2050 at the global for each scenario. Full names for countries and basins are provided in Table A1 in the appendices.**





## 5 Discussion and conclusions

### 5.1 Comparison with existing studies

Previous studies have assessed the cost-effectiveness of adaptation options to address the global impacts of future socioeconomic and climatic changes on water resources, as shown in Table 3. The cost estimates in these studies vary because of differing scenario assumptions, methodologies applied, and input and temporal resolutions, making direct comparisons of outcomes not straightforward. However, despite these differences between our study and other studies, our cost estimates

appear broadly consistent with previous studies. We estimate the costs of water supply and investment in water-related infrastructure (improvement in irrigation efficiency, expansion of non-conventional water supply, expansion of reservoir capacity) at around 642 billion USD in 2050, comparable with estimates provided by Woetzel et al. (2017) and Kirshen (2017). Woetzel et al. (2017) estimated spending on water infrastructure at 200 billion USD in 2016 and 500 billion USD in 2030, whereas Krishen (2017) calculated the cost of water supply production facilities including reservoirs, desalination, and

wastewater treatment over ten regions and found that the total annual adaptation costs amount to 531 billion USD over the period 2000-2030. Strong et al (2020) determined the annual cost for achieving sustainable water management at 1,037 billion USD for the time period 2015-2030. This includes the costs of ensuring universal access to drinking water and sanitation, reducing water pollution and scarcity, and treating industrial wastewater. Our results also show that improving irrigation efficiency might lower annual water costs by 13 billion USD by 2050. This is aligned with the estimates presented by Fischer

et al. (2007), who suggested that by 2080, mitigation through improved irrigation efficiency might lead to annual cost reductions of around 10 billion USD. Lastly, our estimate of the cost of investment in reservoir expansion, improved irrigation efficiency, and non-conventional technologies are around 50 billion USD/year in 2050, which are consistent with the cost estimates of Schmidt-Traub (2015) and Straatsma et al. (2020) to reduce future water gaps and achieve water-related Sustainable Development Goals (SDG6) globally. Straatsma et al. (2020) calculated the investment cost of global annual

adaptation options (improved irrigation practices, increased water supply, and reduced municipal and industrial water use) for SSP2-RCP2.6 in 2090 to be approximately 79 billion USD, while Schmidt-Traub (2015) estimated that 49 billion USD would be needed to ensure access to safe water and improved sanitation. Strong et al. (2020) estimate that implementing supply-side infrastructure solutions to address water scarcity would cost approximately 12 billion USD per year over the 2015–2030 period, whereas Parkinson et al. (2019) found that closing SDG6 infrastructure gaps would require an investment cost of 260 billion

USD per year in 2030, including piped water supply, wastewater collection, and water treatment.





**Table 3: Existing estimates of the adaptation cost of the water sector to future climate and socio-economic scenarios.**

| Study | Objective of the study | Spatial scale | Methodology | Cost estimate |
|---|---|---|---|---|
| Kirshen (2017) | Estimate the cost of water supply production facilities (groundwater, reservoirs, desalination, wastewater treatment) needed by climate and socio-economic changes by 2030. | Over ten regions | Literature review | 531 billion USD over the 2000-2030 period. |
| Fischer et al. (2007) | Assess the water scarcity problem from the perspective of climate change mitigation, estimating the future changes in irrigation efficiency and water costs. | Global | Literature review | Annual cost reductions of about 10 billion USD by 2080 compared to unmitigated scenario. |
| Ward et al. (2010) | Estimates the cost of climate change adaptation for industrial and municipal water. | Global (over 281 water provinces) | Literature review | 12 billion USD/year with 83-90% in developing countries. |
| Straatsma et al. (2020) | Assess the magnitude and the global spatial distribution of the future water gap and determine the cost of adaptation measures in 2090 under the SSP1-RCP2.6 and SSP5-RCP8.5 scenarios. | Global | Literature review | 79 billion USD/year for the SSP1-RCP2.6 scenario and 115 billion USD/year for the SSP5-RCP8.5 scenario in 2090. 36 billion USD/year for Asia, 7 billion USD/year for North America and 6 billion USD/year for Europe in the SSP5-RCP8.5 scenario. |
| | Improved irrigation practices. | | Literature review | Less than 0.2 billion USD/year in North and South America, Africa, Europe and Oceania. 2 (SSP1-RCP2.6) to 3 billion USD/year (SSP5-RCP8.5) in Asia. |
| | Increase water supply (reservoir capacity, desalinated capacity and water reuse). | | Literature review | 28 billion USD/year for SSP5-RCP8.5. 12 billion USD/year for Asia and around 5 for each of Africa, Europe, and North America. |
| | Enhancement in the industrial processes and water saving measures in the domestic sector. | | Literature review | 32 billion USD/year for SSP5-RCP8.5. 17 billion USD/year for Asia, 10 billion USD/year for Africa, 3 billion USD for North America and 2 billion USD/year for Europe in the SSP5-RCP8.5 scenario. |
| Schmidt-Traub (2015) | Determine the investment cost for ensuring access to safe water and improved sanitation, reservoir construction, and flood protection. | Global | Literature review | 49 billion USD/year for the period 2015-2030. |
| Woetzel et al. (2017) | Estimate the current and future spending on water infrastructure (2016-2030). | Global | Literature review | 200 billion USD/year in 2016. 500 billion USD/year in 2030. |
| Parkinson et al (2019) | Estimates the investment costs into water supply and efficiency improvements, closing the SDG6 infrastructure gaps. | Global | Literature review | ~350 billion USD/year in 2030. |
| Strong et al. (2020) | Estimates the cost to deliver sustainable water management (including the costs to access drinking water and sanitation services, reduce water pollution and scarcity, and water management solutions). | Global | Literature review | 1,037 billion USD/year for the time period 2015-2030. |





| | Supply-side infrastructure solutions to breakdown water scarcity such as dams, desalination plants, major basin transfers, and groundwater pumping. | | | 12 billion USD/year. |
|---|---|---|---|---|

## 5.2 New insights from ECHO-Global application

Results indicate that global water withdrawals are expected to rise by 30% by 2050 under the BAU scenario. Results from the application of ECHO-global show that a combination of water management options can help satisfy the demand while minimizing environmental impacts. Demand management options can reduce withdrawals by 27% compared to BAU. Since water scarcity is already a pressing issue in numerous regions of the world, adopting a set of demand management options, including those discussed here, will be essential to limiting withdrawals to sustainable levels. Increases in industrial and

domestic withdrawals are significantly larger than irrigation under all considered future scenarios. Continued economic development in currently low and low-middle income regions of the world, leading to expanding industrial sectors, contributes to higher water demands in these regions. In addition, population growth and urbanization, especially in Sub-Saharan Africa, are the major drivers of increased domestic demand and, subsequently, water use. The ECHO-Global model scenario simulations show potential hotspots for growing industrial and domestic water demands where water management

interventions are most needed.

Management scenarios lead to an overall reduction in water withdrawn for irrigation, but irrigation will continue to hold the largest relative share of total withdrawals and will increase locally in some areas. Efficiency gains are crucial for the overall reduction in most scenarios. Thus, this analysis confirms the notion that global advancements in irrigation efficiency and its monitoring are among the most crucial elements of limiting future increases in water withdrawals in a world with a changing

climate. Rapidly growing populations and their demand for food could potentially lead to relatively high levels of irrigation expansion in Sub-Saharan Africa. In contrast, management options analyzed here suggest reductions in irrigated areas in countries currently applying significant levels of irrigation, such as China. Most significant reductions in irrigation occur for staple crops such as wheat, rice, and maize, while higher-value crops see lower reductions.

While surface water use remains unchanged on average, unregulated groundwater pumping could increase substantially by

160% in BAU by 2050. Management options for reducing non-renewable groundwater pumping are shown to be effective in parts of the world currently facing overexploitation of groundwater resources such as the Ganges, the Arabian Peninsula, or parts of California. Implementing management options for limiting the use of non-renewable groundwater is needed to mitigate the detrimental impacts of its unsustainable use. Locally-adjusted water management interventions for reducing the non-





renewable groundwater pumping can use a mix of demand management options and substitution with alternative sources of
water supply.

Our results also show that the high costs of non-conventional water supply restrict its use to relatively low levels in comparison to other sources of water. Capacity expansion can, however, contribute to an increase by 2050 to more than double the 2010 use levels. The areas showing the highest potential for benefit-maximizing upscaling of desalination or wastewater recycling include arid regions in proximity to coasts and with high population or industry densities, as well as current users of non-
renewable groundwater resources. Even though the net benefits of water only change slightly because the total benefits remain high under all scenarios, costs for water supply increase substantially. Especially in areas with relatively low shares of surface water available to cover relatively high demands, scaling up infrastructure, such as non-conventional water supply and reservoir capacity, can inflate costs. Providing compensations for farmers and industries losing out on revenues due to lower water use is a measure that could help create acceptance for the management options studied here. At the global scale,
mechanisms for sharing the changing benefits of shifting water withdrawal patterns will be essential for achieving economically-profitable and environmentally-sustainable water use.

## 5.3 Outlook and potential future applications

Water quality as an important feature of water scarcity has gained substantial traction in the recent past. HEM applications have started to consider this issue by integrating water quality indicators. For example, water quality management options have
been shown to significantly reduce water scarcity in cost-effective ways in some local areas (Baccour et al., 2024). Future HEMs, including ECHO-Global, will have to increasingly address potential solutions for deteriorating global water quality and its impacts on water scarcity. Groundwater availability is similarly decreasing in several hotspots globally, with many aquifers nearing depletion (Scanlon et al., 2023). While the current ECHO-Global implementation includes groundwater pumping, there is room in this and other HEMs to address the transboundary nature of aquifers and improve the representation
of interactions of groundwater with surface water and ecosystems above ground. Such modeling enhancements will be useful for identifying viable policy and management options for the sustainable use of groundwater resources across borders and basins. Further refinement of groundwater representation in the ECHO-Global is planned in terms of updated data on groundwater availability and pumping costs. Besides groundwater, several transboundary issues can be more adequately addressed in our and other HEMs in the future. Currently, most applications incorporate transboundary cooperation in water
allocation by assessing optimal allocation of water in basins, which commonly span over multiple countries. However, the transboundary aspects of virtual water trade through trade of water embedded in manufactured and agricultural products, as well as the transboundary nature of ecosystem services delivered by water, can be modeled explicitly in the future. Whether





this modeling is best implemented by extending existing models or coupling models such as ECHO-Global with specialized models for issues such as trade must be determined in accordance with the research questions the updated modeling framework

will aim to answer. The underlying process for many of the above potential extensions is represented partially in the hydrological model, which is used for calculating hydrological parameters for applying ECHO-Global. In the future, a more dynamic coupling of the hydrological model CWatM and ECHO-Global will improve the feedback mechanisms of water management options and water availability across water sources, basins, and sectors. Interbasin transfers of water resources are currently implemented in some neighboring basins and could increase in the future where new infrastructure is being

developed. Capturing these transfers in linked hydrological-economic modeling frameworks will be essential for determining their impacts on water resources and economic outcomes.

## 6 Appendices

Appendix A

Table A1 provides a list of river basins and countries included in ECHO-Global.


**Table A1: List of river basins and countries included in ECHO-Global.**

| Basin | Country | Basin | Country | Basin | Country |
|---|---|---|---|---|---|
| Amazon (AMZN) | Brazil (BRA), Central South América (CSA), Colombia (COL), Ecuador (ECU), Peru (PER) | Colorado (COLD) | United States (USA) | Indonesia East (IDNE) | Indonesia (IDN) |
| Amudarja (AMDR) | Afghanistan (AFG), Kazakhstan (KAZ), Tajikistan (TJK), Turkmenistan (TKM), Uzbekistan (UZB) | Columbia (COLM) | Canada (CAN), United States (USA) | Indonesia West (IDNW) | Indonesia (IDN) |
| Amur | China (CHN), Russia (RUS) | Congo (CONG) | Angola (AGO), Central African Republic (CAF), Congo (COG), DRC | Indus (INDS) | China (CHN), India (IND), Pakistan (PAK) |
| Arabian Peninsul (ARBP) | Gulf (GUL), Iraq (IRQ) | Cuba (CUBA) | Caribbean Central America | Ireland (IRLD) | British Isles (VGB) |
| Arkansas (ARKS) | United States (USA) | Danube (DANB) | Adriatic (ADR), Alpine Europe (AEU), Central Europe (CEU), Germany (DEU), Turkey (TUR), Ukraine (UKR) | Italy (ITAL) | Italy (ITA) |
| Baltic (BALT) | Baltic (BAL), Russia (RUS) | Dnieper (DNPR) | Baltic (BAL), Russia (RUS), Ukraine (UKR) | Japan (JAPN) | Japan (JPN) |
| Black Sea (BLAS) | Caucus (CCS), Russia (RUS), Turkey (TUR), Ukraine (UKR) | East African Coa (EAFC) | Burundi (BDI), DRC, Rwanda (RWA), Tanzania (TZA), Uganda (UGA) | Kalahari (KALH) | Botswana (BWA), Namibia (NAM), South Africa (ZAF) |
| Borneo (BORN) | Indonesia (IDN), Malaysia (MYS) | Easten Ghats (EGHT) | India (IND) | Krishna (KRIH) | India (IND) |
| Brahmaputra (BRAP) | Bangladesh (BGD), Bhutan (BTN), China (CHN), India (IND) | Eastern Australia (EAUS) | Australia (AUS) | Lake Balkhash (LBAL) | Kazakhstan (KAZ), Kyrgyzstan (KGZ) |



| Brahmari (BRAM) | India (IND) | Eastern Med (EMED) | Cyprus (CYP), Egypt (EGY), Israel (ISR), Jordan (JOR), Lebanon (LBN), Syria (SYR), Turkey (TUR) | Lake Chad Basin (LCHB) | Cameroon (CMR), Central African Republic (CAF), Chad (TCD), Niger (NER), Nigeria (NGA) |
|---|---|---|---|---|---|
| Britain (BRTN) | British Isles (VGB) | Elbe (ELBE) | Germany (DEU), Scandinavia (SCD) | Langcang Jiang (LANJ) | China (CHN), India (IND) |
| California (CALF) | United States (USA) | Ganges (GANG) | Bangladesh (BGD), China (CHN), India (IND), Nepal (NPL) | Limpopo (LIMP) | Botswana (BWA), Mozambique (MOZ), South Africa (ZAF), Zimbabwe (ZWE) |
| Canada Arctic At (CANA) | Canada (CAN) | Godavari (GODV) | India (IND) | Loire Bordeaux (LBOR) | France (FRA) |
| Caribbean (CARB) | Caribbean Central America (CCA) | Great Basin (GRTB) | United States (USA) | Lower Mongolia (LMNG) | China (CHN), Mongolia (MNG) |
| Cauvery | India (IND) | Great Lakes (GRTL) | Canada (CAN), United States (USA) | Luni (LUNI) | India (IND) |
| Central African (CAFR) | Angola (AGO), Cameroon (CMR), Central African Republic (CAF), Congo (COG), Equatorial Guinea (GIN) (GNQ), Gabon (GAB), Namibia (NAM) | Hail He (HAIH) | China (CHN) | Madagascar (MADG) | Madagascar (MDG) |
| Central America (CAMR) | Caribbean Central America (CCA) | Horn of Africa (HAFR) | Ethiopia (ETH), Kenya (KEN), SoMalia (SOM), Uganda (UGA) | Mahi Tapti (MAHT) | India (IND) |
| Central Australia (CAUS) | Australia (AUS) | Hual He (HUAH) | China (CHN) | Mekong (MEKG) | Myanmar (MMR), Southeast Asia (SAS), Thailand (THA) |
| Central Canada S (CCAN) | Canada (CAN) | Huang He (HUNH) | China (CHN) | Middle Mexico (MDLM) | Mexico (MEX) |
| Chang Jiang (CHJG) | China (CHN) | Iberia East Med (IEMD) | Iberia (IBR) | Mississippi (MSIP) | United States (USA) |
| Chile Coast (CHLC) | Chile (CHL) | Iberia West Atla (IWAT) | Iberia (IBR) | Missouri (MISR) | United States (USA) |
| Chotanagpui (CHTG) | India (IND) | India East Coast (INEC) | India (IND) | Murray Australia (MAUS) | Australia (AUS) |




## 7 Code and data availability

The ECHO-Global model version 1.0 used to conduct the simulations and the input and output data presented in this manuscript
are available from https://zenodo.org/doi/10.5281/zenodo.14391182 (last accessed: 13/12/2024).

## 8 Author contribution

TK developed the model. TK, SB, JJ, RS, PB, JYN, and SA prepared model input data. SB performed the simulations and
associated post-processing. TK, SB, JJ prepared the paper with contributions from RS, PB, JYN, SA, DF, JA, FAW, and YW.
TK acquired the funding and was in charge of project administration and supervision.

## 9 Competing interests

The authors declare that they have no conflict of interest.

## 10 Disclaimer

Any opinions, findings, and conclusions or recommendations expressed in this material do not necessarily reflect the views of
the funding organizations.

## 11 Financial support

This research has received support from the SOS-Water project (Grant Agreement: 101059264) funded under the European
Union's Horizon Europe Research and Innovation Programme.

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
