# Peer review of "Development of the global hydro-economic model (ECHO-Global version 1.0) for assessing the performance of water management options"

_Geoscientific Model Development, 2024_

## Referee Comment (RC2)

In this paper, Kahil et al. present the development of ECHO-Global version 1.0, a global hydro-economic model designed to assess the economic and environmental performance of water management options at the subbasin (BCU) scale. By integrating a detailed representation of water flows from multiple sources—including surface water, groundwater, and non-conventional supplies—with advanced economic benefit functions calibrated via positive mathematical programming, the model simulates future water management scenarios under climate and socio-economic changes (SSP2-RCP6.0). The study offers valuable insights into changes in water withdrawals, irrigated crop areas, and associated economic impacts, and its outputs are broadly consistent with previous global assessments. Although the work represents an important advancement in global water resource modeling, I have several comments (see below).

**Section 2 – Modeling framework**

1. The authors make extensive use of multiple indices (e.g., i, a, u, j, k, w) and a variety of binary or proportional coefficients ($b_i$, $b_a$, etc.) to capture the system's complexity. While this is understandable, clarity would be improved by providing a comprehensive table that summarizes all indices, variables, and coefficients—including their definitions and units.

2. Equation 1 (Headwater Inflow): For BCUs, a critical omission in the headwater inflow representation is the lack of consideration for direct evaporation from natural surface water bodies. While Equation 1 accounts for local runoff and upstream inflows, it doesn't factor in evaporative losses from rivers, lakes, and wetlands prior to reaching the headwater gauge. This can represent substantial water losses, especially in arid regions, leading to significant overestimation of available surface water. While this might be negligible in some BCUs, it becomes critically important in others, for instance, in the Sudd swamp (potentially located in a BCU between Sudan and the Nile polygon), where studies suggest up to 50% of surface water can be lost annually through evaporation. I suggest that the authors either add an evaporation term in Equation 1 or provide a clear justification for why its omission does not affect the overall accuracy of water availability estimates.

3. Equation 4 (Reservoir Storage): The current formulation expresses reservoir dynamics as a net release (outflow minus inflow), which obscures the separate contributions of inflows and outflows and makes tracking mass conservation problematic. A formulation that explicitly distinguishes between inflows, outflows, and evaporation would offer a more transparent representation of reservoir operations.

4. When discussing water stocks, the explicit focus on reservoirs raises the question: what about the natural lakes, aren't they sources of water for irrigation and other purposes in many parts of the globe? which can be substantial water stocks in many regions.

**2.2.3 Groundwater pumping**

5. Although the model distinguishes renewable from non-renewable groundwater pumping, it currently assumes no interaction between groundwater and surface water. Given that groundwater recycling, induced recharge, and lateral exchanges often play pivotal roles in water resource sustainability, especially under climate change scenarios, a more nuanced representation of groundwater-surface water interactions would be highly beneficial. Even if such dynamics are slated for future improvements, acknowledging current limitations and discussing potential impacts on long-term resource sustainability would enhance the study.

**Section 2.2.7 Economics**

6. The coupling of water supply dynamics with economic benefit functions (calibrated via the positive mathematical programming procedure) is strength of the methodology. However, the calibration of these economic functions and the underlying demand and cost parameters inherently introduces uncertainty. The paper would be strengthened by a clearer discussion or sensitivity analysis regarding the impact of parameter uncertainty, especially given that water pricing, elasticity, and sectoral cost estimates play a critical role in determining net benefits. A discussion on how these uncertainties propagate through the model would help evaluate the robustness of the policy simulations

**Section 2.3 – Spatial delineation and node-link network**

7. The pragmatic choice of using BCUs, defined as intersections of river basins with country administrative boundaries, effectively balances computational demands with spatial detail. However, this approach leads to heterogeneous spatial resolution, with some countries (e.g., the USA) represented in much finer detail than others. Such differences may influence the accuracy and policy relevance of country-specific outputs. While I am not suggesting a change to the model's fundamental structure, I recommend that the authors discuss how this heterogeneity might affect results and if there are any plans to explore alternative BCU delineations that could mitigate resolution bias, particularly in regions where aggregated representation may obscure important subnational variability.

**2.4 Model database**

8. Regarding Table 1, the reservoir area-capacity function slope is based on Yigzaw et al. (2018). Notably, there has been some critical review of this dataset—especially regarding the 'area-depth' relationship (see Shrestha et al., 2024, Figure 8). Although area-volume comparisons with other methods appear reasonable, it is unclear how sensitive the model is to potential flaws in this dataset. I recommend that the authors double-check this dataset

and verify whether any uncertainties in the area-depth relationship might have implications for their work.

**Section 3: Water management scenarios**

9. Scenario rationale and uncertainty: The paper presents an extensive set of 2050 water management scenarios under SSP2-RCP6.0. However, the exclusive focus on SSP2-RCP6.0 warrants further justification. Given the uncertainties in water supply prediction alongside those in water demand projections influenced by economic assumptions, the authors should discuss how these uncertainties might affect model outcomes. It would be useful to know whether alternative SSP-RCP combinations were considered or if sensitivity analyses were performed, with these insights ideally integrated into the discussion section.

10. Interdependencies in policy constraints and management strategies: Table 2 outlines detailed policy constraints for the scenarios (BAU, ENV, DM, NC, RES), yet the interplay between supply management and demand management strategies is not fully explained. I recommend that the authors provide a brief clarification on how these various constraints and the associated optimal allocation methods interact. This addition would enhance transparency in how water is allocated among sectors without altering the model's fundamental structure.

---

## Referee Comment (RC3)

The review comments on Kahil et al. "Development of the global hydro-economic model (ECHO-Global version 1.0) for assessing the performance of water management options"

This manuscript presents ECHO-Global, a global hydroeconomic model that integrates physical water flows and economic optimization to assess cross-sectoral water allocation under multiple water management scenarios. The model operates at the scale of 282 basin-country units (BCUs) and captures sector-specific water allocation and use in agriculture, domestic, and industrial sectors, constrained by water availability and infrastructure, while maximizing net economic benefits. The scenario analysis explores supply- and demand-side interventions, including efficiency improvements, land and water reallocation, environmental flow protection, and the use of non-conventional water sources.

The manuscript makes a valuable contribution to the field of global water modeling. ECHO-Global distinguishes itself by combining hydroeconomic balance constraints with economic decision-making across sectors. The application of Positive Mathematical Programming (PMP) to calibrate agricultural land allocation adds credibility to modeled crop choices. Its flexible structure allows for scenario-based assessments aligned with SSP–RCP narratives, providing policy-relevant insights into future water challenges. In this manuscript, the authors demonstrate multiple water management scenarios under the SSP2-RCP6.0 scenario.

The study is timely and addresses an important set of questions regarding the feasibility and trade-offs of water management strategies in the context of climate and socioeconomic change. The model structure is thoughtfully designed, and the manuscript is generally well written and organized.

However, there are some points that I would like the authors to address; Some aspects of the model formulation, scenario design, and interpretation would benefit from further clarification. In particular, additional transparency in how assumptions are specified, how key parameters are derived or constrained, and how results should be interpreted in light of modeling limitations would strengthen the overall contribution. Moreover, the presentation of results could be improved in terms of clarity and consistency, and the structure of the methods section might be adjusted to enhance readability. Some scenario implications may also merit broader contextual discussion.

In summary, this is a promising and ambitious study that contributes to advancing the field of integrated global water resource assessment. With improved clarity in model assumptions, explanation of methods, and framing of results, the manuscript will offer valuable insights to both scientific and policy audiences.

<< Major comments >>

P.6, L140: Although the model equations are defined over time-steps (Section 2.2), monthly water demand inputs are described in Section 2.4.1, and apparently, the total surface water inflow to each BCU is defined as annual value, the model's temporal resolution (e.g., whether it solves monthly or annually) is not explicitly stated. I recommend clearly stating the model time-step and temporal resolution in Section 2.1 or 2.2 to avoid ambiguity.

P.7: While the model includes reservoir evaporation, initial storage levels, and dead storage as components of the reservoir mass balance (e.g., equations 4–7), the description of how these quantities are parameterized remains unclear. For example, evaporation is said to depend on reservoir and climatic features, but no equation or calibration method is provided. Similarly, the sources of initial storage and dead storage values are not described. I recommend that the authors provide additional details on the estimation or data sources for these parameters—particularly for evaporation, which can significantly affect water availability in arid regions.

P.11, L260: While the model optimizes land allocation variables $La_{g,j,k,t}$, it remains unclear whether these represent irrigated land area or total crop area. Given their direct link to irrigation water application, they likely refer to irrigated land, but this should be explicitly clarified in the variable definitions.

P.11, L265-267: While the model adopts a Positive Mathematical Programming (PMP) approach to optimize agricultural land allocation, the manuscript lacks a clear description of how irrigated land areas are derived within this framework. Given the importance of irrigated area changes in explaining scenario results (e.g., reductions in agricultural water withdrawal), a more detailed explanation of the PMP calibration steps—including observed activities, cost functions, and land-use constraints—would greatly improve transparency and reproducibility, especially for readers unfamiliar with PMP.

P.11, L265-267: The current modeling framework optimizes irrigated land area based on economic profitability and water availability, but it does not appear to account for key drivers of future land-use change such as shifts in food demand or climate-induced changes in land suitability (e.g., aridification due to a warmer climate, crop viability under warming). These factors can strongly influence future irrigation patterns, and their exclusion limits the applicability of the model under broader climate–socioeconomic scenarios. I suggest that the authors briefly discuss this limitation, particularly in the context of scenarios (e.g., SSP2–RCP6.0) where food trade is expected to play a key role.

Also, the model appears to omit international trade in agricultural commodities. Since the economic decisions in the model depend on crop profitability at the national level, and since global trade flows can

substantially influence land allocation and irrigation demand, the absence of trade dynamics may limit the realism of the scenario outcomes. I also suggest the authors to discuss this limitation as well briefly.

P,10 Eq20, P.17 L376-376: The model appears to use fixed crop prices based on historical FAOSTAT averages for 2006–2015 (Section 2.4.3), without projecting changes in agricultural commodity prices under future scenarios. Since crop prices are a major driver of land allocation and water use decisions, omitting price projections may limit the ability of the model to reflect plausible economic dynamics under SSP2–RCP6.0 or other scenarios. A discussion of this limitation and its potential effects on model outcomes (i.e., projection uncertainty) would be helpful.

P.12, L291: The objective function of the model uses a discount rate to calculate the net present value of benefits (Eq. 25, Section 2.2.8), but, probably, the specific value used for the discount rate, and its data source or justification, are not mentioned in the manuscript. Given the central role of the discount rate in determining long-term investment and benefit evaluations, I suggest that the authors specify the discount rate used and explain the rationale behind its selection, particularly in relation to standard assumptions in SSP or IAM frameworks.

P.17: While the model considers 13 major irrigated crops, the manuscript does not provide sufficient detail on how crop composition or land allocation changes under different scenarios. Including a summary table or plot showing crop-specific land area shifts would enhance interpretability.

P18, L400: The manuscript states that the ENV scenario minimizes the use of non-renewable groundwater (p.18). However, it remains unclear how this is implemented in the model. Is this achieved via explicit constraints, penalization in the objective function, or higher supply costs? Since this assumption plays a central role in shaping water allocation outcomes under the ENV scenario, it would be helpful if the authors provided more detailed explanation of the modeling formulation underlying this restriction.

P18, L402-403: The manuscript states that the DM scenario "identifies an optimal allocation of water and land to enhance agricultural water use efficiency" (p.18). However, this statement may be somewhat misleading. The optimal allocation of water and land (driven by economic value) primarily serves to maximize economic benefits, not necessarily to increase water use efficiency. Only the direct increase in irrigation efficiency (i.e., reaching the technical maximum in each basin) leads to a clear and quantifiable improvement in agricultural water use efficiency. I suggest rephrasing this sentence to more clearly distinguish between these different mechanisms.

P.19: The manuscript describes the Demand Management (DM) scenario as involving "optimal allocation of water" among sectors based on the economic value of water use. While this formulation (or

expression) may be reasonable within the model, it may appear supply-side oriented to some readers—since it does not directly modify water demand behavior, but rather reallocates supply. I suggest clarifying how this approach qualifies as "demand management" in the context of the scenario narrative, perhaps by distinguishing it from infrastructure expansion or other supply-side interventions. The expression, "Optimal water demand allocation", may be straightforward?

P.19, Table 2: The scenarios DM, NC, and RES assume that irrigation efficiency will be increased to a "maximum efficiency level" for each basin. However, the definition and source of this maximum value remain unclear. It would improve transparency to clarify whether these maximum values are technically feasible (e.g., drip irrigation), economically viable, or derived from empirical benchmarks (e.g., FAO-AQUASTAT or literature-based potential efficiencies). Furthermore, the estimation method and data sources used to determine these maximum efficiency values are not described. Providing such clarification, including potential regional differentiation or reference benchmarks (e.g., FAO-AQUASTAT or literature-based ranges), would greatly improve the transparency and credibility of the scenario assumptions.

P20. L411-413: The manuscript states that the model was both calibrated and validated for the base year 2010 (Chapter 4, first sentence). ① However, it is unclear how the model outputs for year 2010 prior to calibration were computed, and what metrics were used to assess the calibration's effectiveness. ② Moreover, performing both calibration and validation on the same year raises concerns regarding overfitting and the robustness of the model's predictive capacity. I recommend that the authors clarify the calibration procedure and consider including a validation step based on out-of-sample data or a different time period.

P.26, L507: Under the SSP2–RCP6.0 scenario, the reported reduction in irrigated land area warrants further discussion. Does this outcome align with other studies projecting land-use responses under similar scenarios? Including such a discussion would help readers better assess the realism and policy relevance of the model's scenario results.

P20., L409: Broadly, the scenario results are presented without uncertainty ranges, confidence intervals, or sensitivity analyses. Given the strong influence of parameters like willingness to pay, irrigation efficiency, and non-conventional water costs, this deterministic presentation may limit the policy relevance of the results. Including uncertainty bands or conducting a robustness check across plausible parameter ranges would enhance the credibility of the scenarios for decision-makers.

P28, L568-569: The manuscript states that the domestic sector accounts for 55% of gross benefits in 2010, exceeding those of the industrial sector. This result may seem counter-intuitive, as industrial activities typically generate substantial economic outputs per unit of water use. While the manuscript

explains that marginal benefits in domestic use are high for essential needs, the specific parameter values or demand curve assumptions used to generate these results are not clearly shown. I recommend the authors elaborate on the assumptions behind sectoral benefit estimation, especially regarding the benefit functions for the domestic and industrial sectors.

P33, L621: The manuscript states that a combination of water management options can help satisfy water demand. However, under the ENV scenario, both environmental flow requirements and constraints on non-renewable groundwater use are expected to reduce the water available for irrigation. Probably, this leads to substantial decreases in agricultural water withdrawals compared to BAU. It would be helpful for the authors to clarify whether this description is correct and whether these reductions result from unmet water demand due to supply constraints, or from economically optimal decisions under restricted water availability.

P34, L657: While the model provides a detailed representation of water allocation across BCUs within river basins, it does not appear to incorporate institutional or policy-based constraints such as transboundary water treaties or cooperative water management. Similarly, international trade in agricultural products is not modeled, despite its potential impact on regional cropping patterns and water demand. Clarifying these limitations would help define the scope and appropriate applications of the modeling framework.

<< Minor comments >>

P2 L41-46: Would you elaborate or rephase "appropriate water management options … consistent across spatial scales"?

General: I suggest reorganizing Sections 2.2–2.4 so that the spatial delineation and data sources (currently in Sections 2.3 and 2.4) are presented before the model formulation (Section 2.2). This would help readers understand the origins and meaning of key parameters or assumptions—such as willingness to pay and irrigation efficiency—before encountering them in the description on equations. Presenting the data context first would improve the overall readability of the modeling framework.

P.12, Eq. 25: the net present value (NPV) is defined as a summation over time $ttt$, yet the notation "Max NPV" is somewhat ambiguous. It might be clearer to explicitly show the double summation over both $ttt$ and $uuu$, and to define NPV as a function of $ttt$, to clarify that it accumulates time-discounted net benefits across periods.

Figure 3: The colorbar needs unit.

Figure 5(a): Groundwater is shown as a single aggregated category. However, since the model differentiates between renewable and non-renewable groundwater—both conceptually and in terms of cost and sustainability—it would be more informative to distinguish these sources in the figure. This would also better support the interpretation of the ENV scenario, which specifically aims to reduce non-renewable groundwater use. I suggest disaggregating groundwater into renewable and non-renewable components to enhance the clarity and policy relevance of the figure.

Figure 6: Maps appear to be vertically compressed, which may hinder the geographic interpretation of spatial patterns. The aspect ratio does not reflect the natural proportions of the Earth's latitude–longitude grid, making it difficult to compare regions and assess spatial trends accurately. I recommend adjusting the map projection or aspect ratio to improve visual clarity and ensure accurate geographic representation.

---

## Author Response (AR1)

**Response letter to review comments of the manuscript** "Development of the global hydroeconomic model (ECHO-Global version 1.0) for assessing the performance of water management options"

Dear Dr. Kahil and co-authors,

Thanks for your patience with this review process. It took longer than I had hoped, but the reviewers have provided remarkably thoughtful and detailed comments that I hope will prove to be valuable for further enhancing your contribution. In general I agree with the comments from reviewers, which are largely asking for additional details (e.g., more details in the methods regarding model formulation, scenario design, etc., which will help with interpretation of results).

As you prepare your response to reviewer comments, I also wonder whether you have considered developing a dedicated repository (e.g., Github or similar) for Echo-Global (or do you have one and I missed it?), as well as a meta-repository laying out step-by-step instructions for reproducing this specific paper's experiments. Myself, I find it challenging to following how to actually reproduce what you have done in this paper using only the details contained on the zenodo repository, having never used or run this software before. It could be nice to do more hand-holding to maximize reproducibility. I am not suggesting this is a requirement, just raising it as something to think about!

Foremost, we would like to thank the editor and the reviewers for their thoughtful comments that have helped us to improve substantially the quality of the paper. We have carefully addressed all the concerns introduced by the reviewers. The revisions we have made are highlighted in the text and our detailed responses to the reviewers' comments can be found below.

Related to the model reproducibility, we have now amended the Zenodo repository by including more explanation on the different ECHO-Global model version 1.0 components and how the model can be run. We hope this revision will maximize reproducibility.

**Referee #1:**

Tom

**General comments**

I apologize for the time it took to review this paper. This paper includes advanced content covering extensive elements of global water resources. It took much more time than I had initially estimated to review it (I read through it three times).

This paper is ambitious in that it attempts to estimate the amount of water withdrawn and consumed for agricultural, industrial, and domestic sectors for 300 geographical units globally. It calculates the costs and benefits in each unit by water source and maximizes the net benefits for the entire world. While the conventional method of calculating the amount of water withdrawn by sector is to use a regression model, this research proposes a new technique that is suitable for advanced scenario analyses.

The manuscript is well prepared, but I felt that there were issues with the following: in the introduction section, there was no reference to earlier research by GCAM, which has long

propelled economics-based global water resources studies; in the methods section, there was room for improvement in the expression of mathematical formulas; and in the methods, results, and discussion sections, there was no description of the balance between supply and demand of goods and services. For details, please see below.

**Specific comments**

1. Lines 52-68: It is surprising to me that the authors do not refer to any works of GCAM (e.g., Kim et al., 2016; Dolan et al., 2021; Niazi et al., 2024). Niazi et al. (2024) are cited in this manuscript but not in the review part. Their works must be referred to, and this study must be contextualized well in the development history of economics-based global water resources modeling.

Thank you for your comments and providing a list of GCAM references. We are aware of the significant contribution of GCAM-related studies to the economics of global water resources. In our review, we focused exclusively on hydro-economic modeling studies, which are models that represent explicitly water supply and demand nodes and related economics in a spatially explicit way, as described in the review paper of Ortiz-Partida et al. (2023). We understand GCAM as a more integrated assessment model, having a much broader coverage of economic sectors and environmental impacts. However, following the reviewer suggestion and the significant contribution of GCAM-related studies, we have now included relevant GCAM-related studies in our literature review.

2. Line 151 "each headwater gauge": What is the headwater gauge? I assume it is the uppermost river discharge gauge, but Equation 1 has the inflow term (I\_h,t).

The headwater gauge (referred to as the subset h) is a one type of flow nodes (referred to as the set i) that is used to compute water availability in each BCU. Given the size of the BCUs in the ECHO-Global, we assumed that each BCU has one headwater gauge where local runoff and inflow from upstream BCUs are summed up. To avoid confusion, we have replaced the term headwater gauge by the term inflow gauge in the revised version of the manuscript.

3. Line 189 "but it is considered a more expensive water supply source compared to surface water and renewable groundwater": Is this relationship static (irrelevant) to groundwater table depletion?

Groundwater is modelled in a static way in the current version of ECHO-Global, with an estimated cost based on current groundwater table depth and energy costs in each BCU following the same approach used in the study of Kahil et al. (2018). We limit the pumping of renewable groundwater by the natural recharge of aquifers, while non-renewable groundwater is physically unlimited, but assumed to be more expensive than renewable groundwater. However, the marginal cost of pumping groundwater is not increasing with the depletion of groundwater. This approach has been chosen here due to the difficulty of modeling the dynamics of groundwater resources at ECHO-Global spatial scale and the computational burden that might entail. However, this approach allows evaluation of the sustainability of groundwater pumping given the projected use, and simulation of scenarios where maximum monthly renewable groundwater supply is adjusted due to policy interventions or climate change impacts. The inclusion of groundwater dynamics in ECHO-Global might become possible with the release of groundwater datasets such as the one of Niazi et al. (2024, 2025).

4. Line 213 "water application per ha, b\_a,j,k": I expect this is a weather-/climate- dependent term, but the text implies this is a tuning parameter. If this term is not weather-/climate-specific, the projected change in irrigation water requirements due to global warming (e.g., Wada et al. 2013) cannot be implemented in this modeling framework. Elaborate on this issue.

Water application per hectare for each crop depends on climate conditions (i.e., potential evapotranspiration and effective precipitation) and irrigation efficiency. In the section describing ECHO-Global mathematical formulation, we refer to model parameters, which are known coefficients before optimization model solution either they depend on scenario or not, and to model variables, which get known only after the optimization model is being solved. Subsequently in the section model database, we describe the method used to estimate model parameters. To further clarify this, we have now amended the section of mathematical formulation to clearly indicate that irrigation water requirements depend on climate conditions and irrigation efficiency (lines 230-231). We have also changed the name of the coefficient for irrigation water applied, irrigation water consumed, and irrigation water returned from *b* to *w*.

5. Line 265 "alpha\_1,a,g,j,k is...": Do you mean the alpha\_1 is negative? Equations 22 and 23 are expressed in general forms, but I guess the authors assume a declining and a convex upward function, respectively. This should be conveyed more clearly.

We have now revised the section on the computation of crop yields and the PMP calibration procedure. The term  $\alpha_1$  is indeed negative, as the yield function assumes that yields decline as the cultivated area of a given crop expands. Revisions are made in lines 281 to 289.

6. Line 290 Equation 25: Maximizing the sum of net benefit is understandable, but I wonder whether the supply and demand of goods and services equilibrate with this approach. Another concern is what the author assumes on international trade. These points are critically important in interpreting the results part.

Thank you for your thoughtful comment. ECHO maximizes the sum of net benefits from water-related sectors, with the objective to balance the supply and demand of water, but without any constraint on the supply and demand of goods and services. ECHO-Global doesn't include such a constraint for crop production because we are only representing irrigated agriculture and not representing rainfed agriculture, land use competition and international trade. For the industrial sector, ECHO is only modeling its water demand and not the supply and demand of goods and services. Our assumption is that supply of goods and services would be balanced through expanding and intensifying production and international trade. For instance, if the production of a certain crop is reduced in one region of the world, it would be compensated by increasing production in other regions of the world having enough water availability (i.e., not affecting the balance of water in those regions) or lead to increasing market prices. Despite this limitation, due to the versatility of ECHO-Global, additional constraints can be easily added to represent supply or/and demand requirements over space and time as part of scenario or sensitivity analyses. We have now added a paragraph in lines 323-328 to explain this limitation.

**7. Line 307-307 "1410 river gauges nodes and 1128 demand nodes": Clarify why they become 282\*5 and 282\*4.**

We include in ECHO-Global, 5 river gauge nodes in each BCU (282 BCUs \* 5 river gauge nodes = 1410 river gauge nodes). Each BCU has 3 demand nodes (282 BCUs \* 3 demand nodes = 846 demand nodes). The number of demand nodes has been corrected from 1128 to 846, as initially we included in the model 4 demand nodes including livestock demand, which was not considered later.

8. Line 376 "For irrigated agriculture, country-specific prices of 13 crops": Do you mean the market price of agricultural products from irrigated cropland? I wonder whether such a price is distinguishable between rainfed and irrigated agriculture.

Only irrigated agriculture is included in the ECHO-Global. But the market prices of crops (\$ per kg) are the same for irrigated and rainfed agriculture, the difference would be in the crop yields (kg per ha) between irrigated and rainfed agriculture.

9. Line 413 "for the base year 2010. The calibration process consists in adjusting model parameters such as irrigation efficiency...": Was calibration conducted only for one year? If this is the case, the validation should ideally be done for a different year other than 2010. If addressing is technically tricky, at least the authors can discuss challenges.

Thank you for your comments. Differing from biophysical models (e.g., hydrological and crop simulation models), the practice in economic optimization models is to calibrate and validate (or conducting diagnostic tests) for the same year. The calibration approach aims to make the model reproduces satisfactorily the base conditions (or observed conditions) in a representative year (often a year with normal water flow conditions), e.g., crop areas, water use, economic benefits. We often use procedures such as the Positive Mathematical Programming (PMP) to identify parameters in the benefit function that enables reproducing base conditions. The term "positive" implies the use of observed data as part of the model calibration process. As many of the base condition parameters (e.g., crop prices, yields, production costs) would be changing over years, it would be practically impossible to validate an optimization model in different years. The calibrated and validated model would enable us to simulate the effect of changes in policy or climate conditions (through changing parameters or adjusting constraints), but would not enable us forecasting the future. To better explain the calibration procedure of optimization model we refer the readers to the paper by Howitt et al. (2012) about calibrating disaggregate economic models of agricultural production and water management. We have now clearly highlighted this issue in the revised manuscript in the section 4.1 on model validation.

10. Line 524 "the ENV scenario would reduce cereals (wheat, maize, and other cereals) by 36-45%, cotton by 32%, oil crops by 27%, roots by 26%, fruit by 24%, vegetables by 21%, and rice by 14% in 2050": First, which do you mean reduction in the total crop production or irrigation water? If the former is the case, how will humanity be fed by smaller calorie production than today (see discussion in Gerten et al. 2020)? If the latter is the case, would it accompany a considerable expansion in rainfed cropland (see discussion in Rosa et al. 2018)?

We refer to reductions in irrigated areas of those crops. These results are consistent with findings from other global assessments (Popp et al. (2017), Fricko et al. (2017), Gao et al. (2024)), which emphasize the combined effects of increasing water scarcity, intensified competition among sectors, and climate change in constraining irrigated agriculture, especially in already water-stressed regions. Importantly, our results show how ECHO-Global responds to different constraints (or policy) on crop area (proportional land allocation vs. optimal land allocation driven by crop economic value). However, it is important to mention that ECHO-Global does not include the possibility to expand irrigation in basins where converting rainfed areas is possible and water is available. These potential expansions as shown by Rosa et al. (2018) might offset the reductions we are projecting here in our study. Moreover, as indicated by Gerten et al. (2020), constrained irrigation expansion to address environmental objectives would require a transformation towards more sustainable production and consumption patterns to achieve food security, including among others spatial reallocation of crop production, improved waternutrient management, improved water management in rainfed agriculture, food waste reduction and dietary changes. In future work, the ECHO-Global can be further enhanced to address these relevant research and policy questions, by e.g., combining our optimization-based approach with a process-based crop water model similar to the one of Rosa et al. (2018). We have now amended the manuscript to address this very important comment. Changes made in the manuscript are now provided in lines 598 to 608.

11. Line 571 "while the agricultural sector has a smaller share of 7% (23 billion USD/year)": By reading this point, I started to wonder about the role of governments (the public sector). Agricultural water is often subsidized, and its infrastructure is usually built and managed by central/local governments. Does the public sector cover the cost included in Equations 21-22, and how?

Thank you for your thoughtful comment. Our assumption is that the cost of water use from surface water, groundwater and non-conventional water is paid by the users (agriculture, households and industries). The cost of water use can be subsidized, and users might not be paying the full cost of water supply. We also assume that the cost of infrastructure expansion such as building reservoirs is covered by the public sector, and therefore are not included in the model equation (20-24), but their effects are simulated through scenario analysis by changing infrastructure capacity. It is important to note that there is very limited data on water costs and prices, and models such as ECHO-Global could enable simulating scenarios of changes in those parameters and assess their effects on water use decisions. We have added a sentence in lines 431-432 to clarify this issue.

12. Line 601-604 "Strong et al. (2000)...": The discussion sounds blurry. Clarify whether their findings/estimates are similar to (support) the authors' findings or not.

We have addressed this comment as suggested. Please see changes in lines 695-696.

**13. Line 612-615 "Strong et al. (2000)...": Ibid.**

Please see response to previous comment.

**14. Line 622 "Demand management options can reduce withdrawals by 27% compared to BAU": Was goods and service supply maintained? Won't a 27% reduction create other problems?**

Thank you for your comment. This is correct, a reduction of water withdrawals could create other problems, e.g., reduction in the supply of certain commodities. However, as explained in our response to comments # 6 and 10, ECHO-Global doesn't currently consider the balance of supply and demand of other goods and services other than water. For instance, the reduction of the irrigated crop production in certain areas could potentially lead to an increase of rainfed crop production in that area or both rainfed and irrigated crop production in other areas where biophysical and climate conditions allow. It could also lead to a shortage of supply and increase of crop prices. However, this type of assessment is beyond the capability of ECHO-Global and needs to be handled in combination with other models such as GCAM. ECHO was coupled with the agricultural sector model GLOBIOM at basin scale for the Zambezi basin to address similar questions (Please see Palazzo et al. (2024) included now in the references list of the manuscript). In future works, ECHO-Global can incorporate constraints to limit the changes in the production of certain commodities or in certain locations and enabling the expansion of irrigated crop production. We have now added additional text in section 5.3 on limitations of current version of the ECHO-Global model and potential future extensions (lines 744-760).

15. Line 639 "unregulated groundwater pumping could increase substantially by 160% in BAU by 2050": The claim contrasts that of Niazi et al. (2024). If I understand correctly, the difference stems from the treatment/assumption in the groundwater table. If it is explicitly considered, groundwater pumping becomes, sooner or later, economically unaffordable.

The groundwater modeling approach in ECHO-Global is explained in the response to comment #3. It differs from the approach used by Niazi et al. (2024). Based on the results of Niazi et al (2024) (shown in Figure 1), non-renewable groundwater pumping is expected to reach a peak (a median of 625 and a maximum of 1640 km³/year) by 2050, then start declining afterwards till 2100. Our results are limited to 2050 and consider the total amount of groundwater pumping, including renewable and non-renewable groundwater. Our estimate of total groundwater pumping by 2050 ranges from 690-1844 km³/year, depending on scenarios. However, it is difficult to directly compare our results to those of Niazi et al. (2024), given the different assumptions and scenarios considered and considering the limitations of the groundwater representation in ECHO-Global version 1.0. In future works, improvements in the representation of groundwater in ECHO-Global might become possible with the release of groundwater

datasets such as the one of Niazi et al. (2024, 2025). This limitation and future improvements have been now clearly highlighted in the revised manuscript.

**Editorial comments**

- 16. Line 90 Figure 1: The acronyms used (PMP and BCU) should be explained in the caption. Corrected.
- 17. Line 114 "across subbasins within river basins at the global scale": I feel this part is a bit awkward. Subbasins are always within river basins...

Corrected

18. Line 117 "from inverse water demand functions estimated using the Point Expansion approach (Griffin, 2016)": This part is not very informative. I understand that readers should read Griffin (2016), but it would be helpful for us if the authors added more understandable information.

We have now added explanation of the Point Expansion method (lines 126-127). But details of how to estimate the inverse demand function with the Point Expansion method are explained in a more detailed way in Griffin (2016).

19. Line 120 "using the positive mathematical programming (PMP) procedure to address regional-scale aggregation and overspecialization problems (Baccour et al., 2022; Dagnino and Ward, 2012)": ibid.

The PMP approach is now explained in detail in the revised manuscript, also responding to comment #9. Please see new text in lines 290-295.

20. Line 151 "local runoff r\_h,t, and inflow from upstream BCUs I\_h,t": The authors use "X" for flows in general, while here, r and I are also used for "flows (in contrast to stocks)." These expressions are confusing to me, which is one of the reasons why I needed three times of reading.

We agree that it is a bit confusing. In ECHO, we use the set i for any flow node and subsets of i (e.g., r, h) for different types of flow nodes. This way is very helpful for our modeling purposes, and it is one of the powerful features of the GAMS programming language, enabling to reduce significantly the number of equations in the model and facilitate its expansion with additional nodes when needed without having to add equations and variables. The flow variable X for example can be used for any type of flow node and we don't have to create a specific variable for each type of flow node.

21. Line 158 "and -1 for nodes that reduce flow": How streamflow can be reduced? Do you mean "diverted flow?"

In the node link network, we use +1 for any flow node that adds water to the river flow (e.g., inflow from upstream, local runoff, return flows) and -1 for any flow node that reduce the river flow (e.g., water diversion for different purposes).

22. Line 164 Equation 4: This expression is very confusing to me. Here, X is used both to express release and evaporation. It is odd to see that S (Storage) and X(Flow/Flux) have the same dimension (kg).

This equation computes the storage volume at each time step t as a function of the storage volume at time t-1 minus net release (the difference between outflow and inflow from the reservoir) and evaporation (all in m3).

23. Line 222 Equation 14: Why does the form of the equation differ from Equation 12? I expected the difference to be only with/without the irrigation efficiency term.

Equation 12 calculates water applied to crops and equation 14 calculates water used (i.e., consumed) by the crops. Water applied equal water used plus return flows. The equations are written in a different way because we refer to application nodes and use nodes in the node-link network, which are in reality the same.

24. Line 266 "based on the first-order conditions of the agricultural profit maximization problem following the PMP procedure (Dagnino and Ward, 2012)": This phrase is too technical and not informative. Provide a bit more understandable information.

Additional explanations of the PMP procedure were added to the revised manuscript.

**25. Line 268 "M&I": What's this for?**

With M&I we refer to urban (or municipal) and industrial water demand.

26. Line 277 "Then beta\_2,M&I is expected to be large and negative": What does "large" mean? The total benefits equation for domestic and industrial (M&I) use specified in (23) is:

$$TB_{M\&I,t} = \beta_{0,M\&I} + \beta_{1,M\&I} \cdot X_{M\&I,t} + \beta_{2,M\&I} \cdot X_{M\&I,t}^2$$

An expected large and negative  $\beta_{2,M\&l}$  means that total benefits from the first few units of M&l water use rise quickly because of people's high willingness to pay for indoor demand and essential industrial processes. After those demands are satisfied, additional water allocated to the same domestic and industrial use falls off in value quickly. That is, the high point of the quadratic benefits function occurs at a comparatively low level of use, for which that level is calculated by differentiating total benefits with respect to use (X). The benefits maximizing the level of use are  $(\beta_{1,M\&l}$  /-2  $\beta_{2,M\&l}$ ). After that benefit maximizing level of water use occurs, additional use produces a quick reduction in total benefits, since the additional water cannot be used productively for M&I use. In fact, use levels beyond  $(\beta_{1,M\&l}$  /-2  $\beta_{2,M\&l}$ ) will not occur unless the price of water is negative, since at a zero price of water, that total benefit maximizing use level would occur.

**27. Line 397 "a more sustainable scenario (RES)": Maybe it reads ENV.**

The way scenarios are designed in this paper, the RES is the more sustainable one as it incorporates all environmental constraints of the ENV scenario and adds on more adaptation possibilities.

**References**

Kim, S. H., Hejazi, M., Liu, L., Calvin, K., Clarke, L., Edmonds, J., Kyle, P., Patel, P., Wise, M., and Davies, E.: Balancing global water availability and use at basin scale in an integrated assessment model, Climatic Change, 136, 217-231, 10.1007/s10584-016-1604-6, 2016.

Dolan, F., Lamontagne, J., Link, R., Hejazi, M., Reed, P., and Edmonds, J.: Evaluating the economic impact of water scarcity in a changing world, Nature Communications, 12, 1915, 10.1038/s41467-021-22194-0, 2021.

Gerten, D., Heck, V., Jägermeyr, J., Bodirsky, B. L., Fetzer, I., Jalava, M., Kummu, M., Lucht, W., Rockström, J., Schaphoff, S., and Schellnhuber, H. J.: Feeding ten billion people is possible within four terrestrial planetary boundaries, Nature Sustainability, 3, 200-208, 10.1038/s41893-019-0465-1, 2020.

Niazi, H., Wild, T. B., Turner, S. W. D., Graham, N. T., Hejazi, M., Msangi, S., Kim, S., Lamontagne, J. R., and Zhao, M.: Global peak water limit of future groundwater withdrawals, Nature Sustainability, 7, 413-422, 10.1038/s41893-024-01306-w, 2024.

Rosa, L., Rulli, M. C., Davis, K. F., Chiarelli, D. D., Passera, C., and D'Odorico, P.: Closing the yield gap while ensuring water sustainability, Environmental Research Letters, 13, 104002, 10.1088/1748-9326/aadeef, 2018.

Wada, Y., Wisser, D., Eisner, S., Floerke, M., Gerten, D., Haddeland, I., Hanasaki, N., Masaki, Y., Portmann, F. T., Stacke, T., Tessler, Z., and Schewe, J.: Multimodel projections and uncertainties of irrigation water demand under climate change, Geophys. Res. Lett., 40, 4626-4632, 10.1002/grl.50686, 2013.

**Referee #2:**

In this paper, Kahil et al. present the development of ECHO-Global version 1.0, a global hydro-economic model designed to assess the economic and environmental performance of water management options at the subbasin (BCU) scale. By integrating a detailed representation of water flows from multiple sources, including surface water, groundwater, and non-conventional supplies, with advanced economic benefit functions calibrated via positive mathematical programming, the model simulates future water management scenarios under climate and socio-economic changes. The study offers valuable insights into changes in water withdrawals, irrigated crop areas, and associated economic impacts, and its outputs are broadly consistent with previous global assessments. Although the work represents an important advancement in global water resource modeling, I have several comments (see below).

**Section 2 – Modeling framework**

1. The authors make extensive use of multiple indices (e.g., i, a, u, j, k, w) and a variety of binary or proportional coefficients ( $b_i$ ,  $b_a$ , etc.) to capture the system's complexity. While this is understandable, clarity would be improved by providing a comprehensive table that summarizes all indices, variables, and coefficients—including their definitions and units.

Thank you for your helpful suggestion. We have now added the Table A1 explaining all model sets, subsets, parameters and variables in the Appendix.

2. Equation 1 (Headwater Inflow): For BCUs, a critical omission in the headwater inflow representation is the lack of consideration for direct evaporation from natural surface water bodies. While Equation 1 accounts for local runoff and upstream inflows, it doesn't factor in evaporative losses from rivers, lakes, and wetlands prior to reaching the headwater gauge. This can represent substantial water losses, especially in arid regions, leading to significant overestimation of available surface water. While this might be negligible in some BCUs, it becomes critically important in others, for instance, in the Sudd swamp (potentially located in a BCU between Sudan and the Nile polygon), where studies suggest up to 50% of surface water can be lost annually through evaporation. I suggest that the authors either add an evaporation term in Equation 1 or provide a clear justification for why its omission does not affect the overall accuracy of water availability estimates.

Discharge and runoff data are provided by the hydrological model CWatM, which takes into account evaporation losses.

3. Equation 4 (Reservoir Storage): The current formulation expresses reservoir dynamics as a net release (outflow minus inflow), which obscures the separate contributions of inflows and outflows and makes tracking mass conservation problematic. A formulation that explicitly distinguishes between inflows, outflows, and evaporation would offer a more transparent representation of reservoir operations.

All the components of the reservoir water balance equation are properly reflected in the model, but were written in an abstract way due to the way the model code is being written. To ensure consistency between equations in the model code and equations in the manuscript, we suggest keeping the equation as it is, with additional explanation added in the revised manuscript to highlight that the net release is the difference between outflow from the reservoir and inflow to the reservoir.

4. When discussing water stocks, the explicit focus on reservoirs raises the question: what about the natural lakes, aren't they sources of water for irrigation and other purposes in many parts of the globe? which can be substantial water stocks in many regions.

We agree with this comment. However, even the most advanced global hydrological models do not include natural lakes. It is also challenging to identify how those lakes are being operated. At this stage, we include only man-made reservoirs included in the GRAND database. The inclusion of natural lakes in ECHO-Global might become possible when required input data become available.

**2.2.3 Groundwater pumping**

5. Although the model distinguishes renewable from non-renewable groundwater pumping, it currently assumes no interaction between groundwater and surface water. Given that groundwater recycling, induced recharge, and lateral exchanges often play pivotal roles in water resource sustainability, especially under climate change scenarios, a more nuanced representation of groundwater-surface water interactions would be highly beneficial. Even if such dynamics are slated for future improvements, acknowledging current limitations and discussing potential impacts on long-term resource sustainability would enhance the study.

Groundwater is modelled in a static way in the current version of ECHO, with an estimated cost based on current groundwater table depth and energy costs in each BCU following the same approach used in the study of Kahil (2018). We limit the pumping of renewable groundwater by the natural recharge of aquifers, while non-renewable groundwater is physically unlimited, but assumed to be more expensive than renewable groundwater. However, the marginal cost of pumping groundwater is not increasing with the depletion of groundwater. This approach has been chosen here due to the difficulty of modeling the dynamics of groundwater resources at ECHO spatial scale and the computational burden that might entail. However, this approach allows evaluation of the sustainability of groundwater pumping given the projected use, and simulation of scenarios where maximum monthly renewable groundwater supply is adjusted due to policy interventions or climate change impacts. The inclusion of groundwater dynamics in ECHO-Global might become possible with the release of groundwater datasets such as the one of Niazi et al. (2024, 2025). We have now highlighted in the revised manuscript this limitation in lines 784-788.

**Section 2.2.7 Economics**

6. The coupling of water supply dynamics with economic benefit functions (calibrated via the positive mathematical programming procedure) is strength of the methodology. However, the calibration of these economic functions and the underlying demand and cost parameters inherently introduces uncertainty. The paper would be strengthened by a clearer discussion or sensitivity analysis regarding the impact of parameter uncertainty, especially given that water

pricing, elasticity, and sectoral cost estimates play a critical role in determining net benefits. A discussion on how these uncertainties propagate through the model would help evaluate the robustness of the policy simulations.

Thank you for your thoughtful comment. We fully agree that many assumptions on model parameters have been made to calibrate the model, which introduces uncertainty into the model. Our approach to address this issue was to run several scenarios as shown in Table 2, in which constraints have been added progressively to check how the model responds to those changes and whether model responses make sense. Despite the uncertainty, ECHO-Global could currently serve well for a comparative analysis of the effects of different

scenarios and management options, since the errors and limitations are applied uniformly across model runs. Running some additional sensitivity tests would be possible, but given the large-scale nature of the model and the many parameters included, we believe the results of the sensitivity tests would not help to better understand the model. We have clearly highlighted the issue of uncertainty in section 5.3 on limitations and future works.

**Section 2.3 – Spatial delineation and node-link network**

7. The pragmatic choice of using BCUs, defined as intersections of river basins with country administrative boundaries, effectively balances computational demands with spatial detail. However, this approach leads to heterogeneous spatial resolution, with some countries (e.g., the USA) represented in much finer detail than others. Such differences may influence the accuracy and policy relevance of country-specific outputs. While I am not suggesting a change to the model's fundamental structure, I recommend that the authors discuss how this heterogeneity might affect results and if there are any plans to explore alternative BCU delineations that could mitigate resolution bias, particularly in regions where aggregated representation may obscure important subnational variability.

The current spatial delineation was chosen to be consistent with other global models and assessments such as the IFPRI's IMPACT-WATER model, but also to reduce the computational burden of the model. Our understanding is that this delineation was created to highlight the most important river basins and countries in terms of water use, especially for irrigation purposes. It has obviously several limitations, one of which is the different level of spatial details covered in different countries and the homogeneity assumption of the BCUs (i.e., each BCU is treated as a single unit, meaning that water flows between spatial locations within a BCU are not considered and water use is aggregated). However, one advantage of the ECHO model is its versatility, enabling to adjust the spatial delineation depending on the research questions. So, when more spatial details would be needed for a certain country, the delineation can be adapted to the need, while model sets, variables and equations should not (or only slightly) change. Model parameters will need to be adjusted to include the input data for the newly added spatial units. We have added the limitations of this approach in section 2.3 (Lines 339-346).

**2.4 Model database**

8. Regarding Table 1, the reservoir area-capacity function slope is based on Yigzaw et al. (2018). Notably, there has been some critical review of this dataset—especially regarding the 'area-depth' relationship (see Shrestha et al., 2024, Figure 8). Although area-volume comparisons with other methods appear reasonable, it is unclear how sensitive the model is to potential flaws in this dataset. I recommend that the authors double-check this dataset and verify whether any uncertainties in the area-depth relationship might have implications for their work.

Thank you for your comment and for referring us to the paper of Shrestha et al. (2024). While we understand the concerns regarding the dataset of Yigzaw et al. (2018). However, at the time of preparing the input data for ECHO-Global, that dataset was the best available. We believe that ECHO is not so sensitive to those reservoir parameters, especially due to the aggregation of

individual reservoir information into the BCU level. In the revised version of the manuscript in the section 2.4.2, we now mention the limitation of the dataset.

**Section 3: Water management scenarios**

9. Scenario rationale and uncertainty: The paper presents an extensive set of 2050 water management scenarios under SSP2-RCP6.0. However, the exclusive focus on SSP2-RCP6.0 warrants further justification. Given the uncertainties in water supply prediction alongside those in water demand projections influenced by economic assumptions, the authors should discuss how these uncertainties might affect model outcomes. It would be useful to know whether alternative SSP-RCP combinations were considered or if sensitivity analyses were performed, with these insights ideally integrated into the discussion section.

We thank the reviewer for this thoughtful observation. The focus of this study is to assess the sensitivity of model outcomes to changes in key water management parameters, using SSP2-RCP6.0 as a baseline scenario for future climatic and socio-economic changes. However, we agree that in future work, going beyond model description, a wider range of scenarios would need to be considered. We have now clearly highlighted this issue in section 5.3 on limitations and future work.

10. Interdependencies in policy constraints and management strategies: Table 2 outlines detailed policy constraints for the scenarios (BAU, ENV, DM, NC, RES), yet the interplay between supply management and demand management strategies is not fully explained. I recommend that the authors provide a brief clarification on how these various constraints and the associated optimal allocation methods interact. This addition would enhance transparency in how water is allocated among sectors without altering the model's fundamental structure.

We appreciate the reviewer's suggestion. Clarifications on the interactions between supply-side and demand-side management strategies, and how they jointly influence the optimal water allocation, have been clarified and new text added to the revised manuscript in lines 459-464.

**Referee #3:**

The review comments on Kahil et al. "Development of the global hydro-economic model (ECHO-Global version 1.0) for assessing the performance of water management options"

This manuscript presents ECHO-Global, a global hydroeconomic model that integrates physical water flows and economic optimization to assess cross-sectoral water allocation under multiple water management scenarios. The model operates at the scale of 282 basin-country units (BCUs) and captures sector-specific water allocation and use in agriculture, domestic, and industrial sectors, constrained by water availability and infrastructure, while maximizing net economic benefits. The scenario analysis explores supply- and demand-side interventions, including efficiency improvements, land and water reallocation, environmental flow protection, and the use of non-conventional water sources.

The manuscript makes a valuable contribution to the field of global water modeling. ECHO-Global distinguishes itself by combining hydroeconomic balance constraints with economic decision-making across sectors. The application of Positive Mathematical Programming (PMP) to calibrate agricultural land allocation adds credibility to modeled crop choices. Its flexible structure allows for scenario-based assessments aligned with SSP–RCP narratives, providing policy-relevant insights into future water challenges. In this manuscript, the authors demonstrate multiple water management scenarios under the SSP2-RCP6.0 scenario.

The study is timely and addresses an important set of questions regarding the feasibility and trade-offs of water management strategies in the context of climate and socioeconomic change. The model structure is thoughtfully designed, and the manuscript is generally well written and organized.

However, there are some points that I would like the authors to address; Some aspects of the model formulation, scenario design, and interpretation would benefit from further clarification. In particular, additional transparency in how assumptions are specified, how key parameters are derived or constrained, and how results should be interpreted in light of modeling limitations would strengthen the overall contribution. Moreover, the presentation of results could be improved in terms of clarity and consistency, and the structure of the methods section might be adjusted to enhance readability. Some scenario implications may also merit broader contextual discussion.

In summary, this is a promising and ambitious study that contributes to advancing the field of integrated global water resource assessment. With improved clarity in model assumptions, explanation of methods, and framing of results, the manuscript will offer valuable insights to both scientific and policy audiences.

Thank you for your thoughtful review and constructive feedback on our manuscript. We sincerely appreciate your recognition of the model's novelty and policy relevance. In response to your suggestions, we have carefully revised the manuscript to improve its clarity, transparency, and overall impact.

**Major comments**

1. P.6, L140: Although the model equations are defined over time-steps (Section 2.2), monthly water demand inputs are described in Section 2.4.1, and apparently, the total surface water inflow to each BCU is defined as annual value, the model's temporal resolution (e.g., whether it solves monthly or annually) is not explicitly stated. I recommend clearly stating the model time-step and temporal resolution in Section 2.1 or 2.2 to avoid ambiguity.

The ECHO-Global version 1.0 model operates at a monthly time step and is run for the years 2010 and 2050. To ensure clarity, we have explicitly stated the model's temporal resolution in Section 2.1 (line 139).

2. P.7: While the model includes reservoir evaporation, initial storage levels, and dead storage as components of the reservoir mass balance (e.g., equations 4–7), the description of how these quantities are parameterized remains unclear. For example, evaporation is said to depend on reservoir and climatic features, but no equation or calibration method is provided. Similarly, the sources of initial storage and dead storage values are not described. I recommend that the authors provide additional details on the estimation or data sources for these parameters—particularly for evaporation, which can significantly affect water availability in arid regions.

We have now clarified the parameterization of key components of the reservoir mass balance in section 2.2.1 (Lines: 182-185). Specifically, evaporation rate is derived from CWatM model simulations (Burek et al., 2020), as the average from 4 climate models: GFDL-ESM2M, HadGEM2-ES, IPSL-CM5A-LR, MIROC5. The initial reservoir storage level is assumed to be 50% of the reservoir's total capacity. Regarding dead storage, it is set to zero in the current version of the model, reflecting a simplified assumption that all stored water is potentially available for use.

3. P.11, L260: While the model optimizes land allocation variables Lag,j,k,t, it remains unclear whether these represent irrigated land area or total crop area. Given their direct link to irrigation water application, they likely refer to irrigated land, but this should be explicitly clarified in the variable definitions.

We have now included the definition of the variable  $L_{ag,j,k,t}$  in Section 2.2.7 (Lines: 279-280), which refers to the irrigated land area for crop j, irrigation technology k, and time t.

4. P.11, L265-267: While the model adopts a Positive Mathematical Programming (PMP) approach to optimize agricultural land allocation, the manuscript lacks a clear description of how irrigated land areas are derived within this framework. Given the importance of irrigated area changes in explaining scenario results (e.g., reductions in agricultural water withdrawal), a more detailed explanation of the PMP calibration steps—including observed activities, cost functions, and land-use constraints—would greatly improve transparency and reproducibility, especially for readers unfamiliar with PMP.

Thank you for your thoughtful comment. In the ECHO-Global model, irrigated land areas are endogenously determined through an optimization that maximizes net benefits across agricultural activities for each BCU, subject to water and land availability constraints.

The PMP procedure allows us to calibrate the model so that the base-year observed land and water allocations are correctly reproduced, addressing the issue of agricultural overspecialization (Howitt, 1995). As detailed in the model validation section (4.1), following calibration, we performed a validation step (or better refer to it as conducting diagnostic tests) comparing observed data and simulated model outputs, including water use, irrigated area, and agricultural income, to ensure that our model accurately reflects real-world conditions and can be reliably used for future scenario simulations.

The idea of PMP was originally introduced by Howitt (1995) to calibrate the solutions of agricultural optimization models. Howitt's original approach uses a cost function calibration, while Dagnino and Ward (2012) use a yield function calibration, enabling a more realistic representation of observed land use and resource allocation.

In our case, the PMP procedure for agricultural activities follows the approach developed by Dagnino and Ward (2012), which uses first-order conditions for profit maximization to estimate the parameters of a crop-specific quadratic production function, thereby replicating observed production data. In such a way, the optimal production solution by our model is close to observed production (calibration).

The PMP calibration procedure begins by collecting observed data for the reference year, including land allocation, crop yields, prices, and crop-non-water production costs, as shown in the economic data of Table 1 in the manuscript. The model then formulates an objective function that maximizes the total net benefits across all crops, subject to land and water resource constraints. The land constraint ensures that the optimal land allocated to each crop and technology does not exceed the observed area. The water constraint ensures that agricultural water use does not exceed the water availability in the system (Eqs. 8, 9, 10 in the manuscript).

The Ricardian rent principle is applied in the yield function by assuming that yields decline as the cultivated area of a given crop expands. In other words, the most productive land is allocated first, and additional land brought into production is progressively less suitable, leading to lower average yields. While this approach simplifies the factors contributing to yield decline, it effectively captures key aspects of farmers' behavioral responses. The yield function is linear and decreasing in the irrigated area of the crop.  $Y = \alpha_0 + \alpha_1 \cdot L$  (Eq. 22). Here,  $\alpha_0$  represents the yield on the first unit of irrigated land area cultivated, while  $\alpha_1$  captures the marginal effect of additional irrigated land area on average yield, with  $\alpha_1 < 0$ .

As crop production (P) is defined as the product of yield (Y) and crop irrigated land area (L), the resulting production function is quadratic in land:

```
\Rightarrow P= Y·L = (\alpha_0 + \alpha_1 \cdot L) \cdot L = \alpha_0 \cdot L + \alpha_1 \cdot L^2
```

We have now included a detailed explanation regarding the PMP calibration in Section 2.2.7 of the revised manuscript (Lines 281-295).

5. P.11, L265-267: The current modeling framework optimizes irrigated land area based on economic profitability and water availability, but it does not appear to account for key drivers of future land-use change such as shifts in food demand or climate-induced changes in land suitability (e.g., aridification due to a warmer climate, crop viability under warming). These factors can strongly influence future irrigation patterns, and their exclusion limits the applicability of the model under broader climate—socioeconomic scenarios. I suggest that the authors briefly discuss this limitation, particularly in the context of scenarios (e.g., SSP2—RCP6.0) where food trade is expected to play a key role.

Yes, you are right. The limitations related to the inclusion of future land-use change and food trade in the model are addressed in the discussion section. It is important to mention that ECHO-Global doesn't allow projecting changes in irrigated crop area, but rather simulates the effect of changes in water availability or other constraints and incentives (e.g., changing water prices, providing subsidies for investing in efficient irrigation systems) on current irrigated crop mix. In future work, the ECHO-Global can be further enhanced to consider land use change and food trade, either through coupling with other sectoral models or incorporating additional constraints to e.g., enable the expansion of irrigated crop production. We have now added additional text in section 5.3 on limitations of current ECHO-Global model version 1.0 and potential future extensions (lines 764-767).

6. Also, the model appears to omit international trade in agricultural commodities. Since the economic decisions in the model depend on crop profitability at the national level, and since global trade flows can substantially influence land allocation and irrigation demand, the absence of trade dynamics may limit the realism of the scenario outcomes. I also suggest the authors to discuss this limitation as well briefly.

As explained in the response to the previous comment, the current version of Global-ECHO doesn't consider international trade. We further highlighted this limitation as well as potential future development of the model in the discussion section (lines 769-772).

7. P,10 Eq20, P.17 L376-376: The model appears to use fixed crop prices based on historical FAOSTAT averages for 2006–2015 (Section 2.4.3), without projecting changes in agricultural commodity prices under future scenarios. Since crop prices are a major driver of land allocation and water use decisions, omitting price projections may limit the ability of the model to reflect plausible economic dynamics under SSP2–RCP6.0 or other scenarios. A discussion of this limitation and its potential effects on model outcomes (i.e., projection uncertainty) would be helpful.

A discussion of crop price projections is included in the discussion section (Lines 772-774).

8. P.12, L291: The objective function of the model uses a discount rate to calculate the net present value of benefits (Eq. 25, Section 2.2.8), but, probably, the specific value used for the discount rate, and its data source or justification, are not mentioned in the manuscript. Given the central role of the discount rate in determining long-term investment and benefit evaluations, I suggest that the authors specify the discount rate used and explain the rationale behind its selection, particularly in relation to standard assumptions in SSP or IAM frameworks. Thank you for your comment. The discount rate is included in the equation of the objective

function. However, it was set at 0, given that we run this version of ECHO-Global in a static mode

(for 1 year) and not in an intertemporal mode (for many years). In this paper, we are interested in analyzing the change in model outcomes when a model parameter changes, comparing the outcomes with and without the change. This way helps identifying model sensitivity to changes in parameters, avoiding the complexity of solving a dynamic version of the model.

9. P.17: While the model considers 13 major irrigated crops, the manuscript does not provide sufficient detail on how crop composition or land allocation changes under different scenarios. Including a summary table or plot showing crop-specific land area shifts would enhance interpretability.

Irrigated cropland distribution at the global scale in 2010 and 2050 for each scenario is already presented in Figure 7a. Moreover, detailed results are provided in the output files in the Zenodo repository [https://zenodo.org/doi/10.5281/zenodo.14391182].

10. P18, L400: The manuscript states that the ENV scenario minimizes the use of non-renewable groundwater (p.18). However, it remains unclear how this is implemented in the model. Is this achieved via explicit constraints, penalization in the objective function, or higher supply costs? Since this assumption plays a central role in shaping water allocation outcomes under the ENV scenario, it would be helpful if the authors provided more detailed explanation of the modeling formulation underlying this restriction.

The ENV scenario minimizes the use of non-renewable groundwater by incorporating a specific constraint into the model. For most BCUs, non-renewable groundwater use is set to zero, reflecting a policy of groundwater sustainability and minimizing reliance on these limited resources. However, in a few BCUs where non-renewable groundwater is essential for domestic water supply and to maintain model feasibility, limited use of non-renewable groundwater was allowed. This approach ensures that non-renewable groundwater is only utilized where necessary, aligning with best practices for sustainable groundwater management. We have now included more details in section 3 of the revised manuscript (Lines 447-449).

11. P18, L402-403: The manuscript states that the DM scenario "identifies an optimal allocation of water and land to enhance agricultural water use efficiency" (p.18). However, this statement may be somewhat misleading. The optimal allocation of water and land (driven by economic value) primarily serves to maximize economic benefits, not necessarily to increase water use efficiency. Only the direct increase in irrigation efficiency (i.e., reaching the technical maximum in each basin) leads to a clear and quantifiable improvement in agricultural water use efficiency. I suggest rephrasing this sentence to more clearly distinguish between these different mechanisms.

In the DM scenario, efficiency is improved in two key ways: First, technical efficiency is enhanced by reaching the technical maximum irrigation efficiency in each BCU; and second, water is also allocated based on economic value, ensuring that water is allocated to uses with the highest marginal returns, thereby enhancing economic efficiency. This dual approach not only reduces physical losses but also optimizes the economic productivity of water resources.

The sentence has been rephrased for clarity as: "The demand management (DM) scenario identifies an optimal water and land allocation that enhances the economic productivity of water across sectors, while also improving technical irrigation efficiency."

12. P.19: The manuscript describes the Demand Management (DM) scenario as involving "optimal allocation of water" among sectors based on the economic value of water use. While this formulation (or expression) may be reasonable within the model, it may appear supply-side oriented to some readers—since it does not directly modify water demand behavior, but rather reallocates supply. I suggest clarifying how this approach qualifies as "demand management" in the context of the scenario narrative, perhaps by distinguishing it from infrastructure expansion

or other supply-side interventions. The expression, "Optimal water demand allocation", may be straightforward?

The concept of "demand management" can be interpreted in various ways. In our modeling framework, the ENV scenario prioritizes water allocation to urban and industrial sectors over agriculture, while the Demand Management (DM) scenario assumes equal access across sectors, without pre-assigned priorities. As previously noted, demand management here refers to the optimization of water allocation among sectors based on economic value, which would enhance economic water productivity without increasing water use (or in other words, better use of limited water resources). This approach captures water allocation patterns driven by relative economic returns, reflecting behavioral adaptations aligned with the core principles of demand management: improving efficiency, promoting sustainability, and maximizing the value of limited water resources.

In response to the reviewer's suggestion, we have revised the expression "optimal allocation of water" to "optimal water demand allocation" and improved the description of the DM scenario for greater clarity and consistency (see Lines 449–451).

13. P.19, Table 2: The scenarios DM, NC, and RES assume that irrigation efficiency will be increased to a "maximum efficiency level" for each basin. However, the definition and source of this maximum value remain unclear. It would improve transparency to clarify whether these maximum values are technically feasible (e.g., drip irrigation), economically viable, or derived from empirical benchmarks (e.g., FAO-AQUASTAT or literature-based potential efficiencies). Furthermore, the estimation method and data sources used to determine these maximum efficiency values are not described. Providing such clarification, including potential regional differentiation or reference benchmarks (e.g., FAO-AQUASTAT or literature-based ranges), would greatly improve the transparency and credibility of the scenario assumptions.

Current levels of irrigation efficiency at country level are obtained from FAO-AQUASTAT database. BCUs are assigned their corresponding country efficiency value. In the DM, NC, and RES scenarios, irrigation efficiency at the BCU level was assumed to increase to the current highest value within their basin. This method ensures that efficiency increases are potentially achievable and consistent with the regional context. Clarifications have been included in Lines 449-455.

14. P20. L411-413: The manuscript states that the model was both calibrated and validated for the base year 2010 (Chapter 4, first sentence). ① However, it is unclear how the model outputs for year 2010 prior to calibration were computed, and what metrics were used to assess the calibration's effectiveness. ② Moreover, performing both calibration and validation on the same year raises concerns regarding overfitting and the robustness of the model's predictive capacity. I recommend that the authors clarify the calibration procedure and consider including a validation step based on out-of-sample data or a different time period.

Thank you for your thoughtful comment. However, we would like to clarify that the calibration and validation processes in hydro-economic models differ significantly from those applied in biophysical models (e.g., hydrological and crop simulation models). The practice in economic optimization models is to calibrate and validate (or conducting diagnostic tests) for the same year. The calibration approach aims to make the model reproduces satisfactorily the base conditions (or observed conditions) in a representative year (often a year with normal water flow conditions), e.g., crop areas, water use, economic benefits. We often use procedures such as the Positive Mathematical Programming (PMP) to identify parameters in the benefit function that enables reproducing base conditions. The term "positive" implies the use of observed data as part of the model calibration process. As many of the base condition parameters (e.g., crop prices, yields, production costs) would be changing over years, it would be practically impossible to validate an optimization model in different years. The calibrated and validated model would

enable us to simulate the effect of changes in policy or climate conditions (through changing parameters or adjusting constraints), but would not enable us forecasting future economic outcomes. To better explain the calibration procedure of optimization model we refer the readers to the paper by Howitt et al. (2012) about calibrating disaggregate economic models of agricultural production and water management. We have now clearly highlighted this issue in the revised manuscript in the section 4.1 on model validation.

**Model Outputs for 2010 Prior to Calibration:**

A pre-calibration run was first conducted using observed hydrological inputs (inflows, groundwater recharge, reservoir capacity, and technological capacities such as non-conventional water supply and crop water requirements) and economic data (crop irrigated areas, water and crop prices, production costs, etc.). This "raw" simulation served only as a starting point; its outputs were not used for analysis but allowed us to quantify the adjustments required during calibration.

**Calibration Process:**

The model underwent a **two-step calibration process**: hydrological and economic. First, the **hydrological calibration** aimed to replicate the observed 2010 water allocation across sectors (domestic, industrial, agricultural) and sources (surface, groundwater, and non-conventional) for each Basin-Country Unit (BCU). The hydrological calibration involves introducing slack variables to represent unmeasured water sources or uses, such as upstream demands, open water evaporation, percolation, and groundwater discharge, to ensure a balance between water supply and demand at each node within the BCU. Calibration is conducted iteratively from upstream to downstream nodes for each BCU, and once an adequate fit replicating observed data is achieved, the unmeasured components are held constant to isolate the effects of policy scenarios. Second, we applied Positive Mathematical Programming (PMP) for **economic calibration** to reproduce observed crop mix, water use and economic benefits in the reference year (for further details regarding the PMP calibration, see our response to question #4).

We assessed **calibration effectiveness** by comparing observed and simulated values at the BCU level using goodness-of-fit metrics: the Nash–Sutcliffe Efficiency (NSE) for hydrological parameters and the PMP itself for economic data. The Nash–Sutcliffe Efficiency (NSE) evaluates how well simulated water use matches observed values, with values closer to 1 indicating a better model fit and higher predictive accuracy.

After calibrating at the BCU level, we conducted a **validation step** to verify that water use by sector and source, irrigated area by crop type, and agricultural income accurately reflect observed data at the country and global levels.

Regarding concerns about **potential overfitting** from performing both calibration and validation in the same year, it is important to highlight that our model operates across multiple spatial scales (BCU, basin, country, and global). The goal of validation is not to fit the data perfectly but to ensure the model captures realistic system behavior under specific constraints for the different scales, thereby maintaining generalizability within the context studied.

We have included further details regarding the calibration and validation procedure in the revised manuscript lines 189-191, 288-295, and 474-478.

15. P.26, L507: Under the SSP2—RCP6.0 scenario, the reported reduction in irrigated land area warrants further discussion. Does this outcome align with other studies projecting land-use responses under similar scenarios? Including such a discussion would help readers better assess the realism and policy relevance of the model's scenario results.

We thank the reviewer for this valuable comment. We discussed the decrease in irrigated area in Lines 598-608.

Lines 598-608: "The reduction in irrigated land area projected in our study aligns with findings from global assessments, which emphasize the combined effects of increasing water scarcity, intensified competition among sectors, and climate change in constraining irrigated agriculture, especially in already water-stressed regions. Popp et al. (2017) and Fricko et al. (2017) highlight that rising water stress, driven by both socioeconomic developments and climate pressures, limits the expansion of irrigation and, in some models, results in stagnation or even regional declines in irrigated areas. These studies further show that while total cropland may continue to grow in the SSP2, the share that can be irrigated is increasingly restricted by limited water availability and growing intersectoral demands. Supporting this trend, Gao et al. (2024) project a global decline in the area equipped for irrigation by approximately 9.4% under SSP2 and 7.1% under SSP1 between 2020 and 2100. Rosa et al. (2018), however, show using a process-based crop water model that a sustainable irrigation expansion and intensification would still be possible, enabling a 24% increase in calorie production. It is important to note that the current version of ECHO-Global does not allow irrigated crop land expansion."

**References:**

- Popp, A., Calvin, K., Fujimori, S., Havlik, P., Humpenöder, F., Stehfest, E., ... & van Vuuren,
  D. P. (2017). Land-use futures in the shared socio-economic pathways. Global Environmental Change, 42, 331-345.
- Yulian Gao, Kecui Dong, Yaojie Yue. Projecting global fertilizer consumption under shared socioeconomic pathway (SSP) scenarios using an approach of ensemble machine learning. Science of The Total Environment. 2024,912,169130. DOI:10.1016/j.scitotenv.2023.169130
- Fricko, O., Havlik, P., Rogelj, J., Klimont, Z., Gusti, M., Johnson, N., ... & Riahi, K. (2017).
  The marker quantification of the Shared Socioeconomic Pathway 2: A middle-of-the-road scenario for the 21st century. Global Environmental Change, 42, 251-267.
- Rosa L, Rulli MC, Davis KF, Chiarelli DD, Passera C, D'Odorico P. Closing the yield gap while ensuring water sustainability. Environmental Research Letters. 2018 Sep 24;13(10):104002.

16. P20., L409: Broadly, the scenario results are presented without uncertainty ranges, confidence intervals, or sensitivity analyses. Given the strong influence of parameters like willingness to pay, irrigation efficiency, and non-conventional water costs, this deterministic presentation may limit the policy relevance of the results. Including uncertainty bands or conducting a robustness check across plausible parameter ranges would enhance the credibility of the scenarios for decision-makers.

Thank you for your thoughtful comment. We fully agree that many assumptions on model parameters introduce uncertainty into the model. Our approach to address this issue was to run several scenarios as shown in Table 2, in which constraints have been added progressively to check how the model responds to those changes and whether model responses make any sense. Despite the uncertainty, ECHO-Global could currently serve well for a comparative analysis of the effects of different scenarios and management options, since the errors and limitations are applied uniformly across model runs. Running some additional sensitivity tests would be possible, but given the large scale nature of the model and the many parameters included, we believe the results of the sensitivity tests would not help to better understand the model and could lead to infeasibilities. We have clearly highlighted the issue of uncertainty in section 5.3 on limitations and future works.

17. P28, L568-569: The manuscript states that the domestic sector accounts for 55% of gross benefits in 2010, exceeding those of the industrial sector. This result may seem counterintuitive, as industrial activities typically generate substantial economic outputs per unit of water use. While the manuscript explains that marginal benefits in domestic use are high for

essential needs, the specific parameter values or demand curve assumptions used to generate these results are not clearly shown. I recommend the authors elaborate on the assumptions behind sectoral benefit estimation, especially regarding the benefit functions for the domestic and industrial sectors.

Thank you for your comment. It is important to mention that ECHO-Global computes the total economic value of water (not the value added of the sector, which measures the economic contribution of a sector to the overall economy as the value of output minus intermediate consumption) in the domestic and industrial sectors measured by the economic surplus, defined as consumer plus producer surplus, derived from the inverse water demand functions. In most industries (except hydropower), water contributes a small portion to the value of the output. Urban water used for essential purposes such as drinking and sanitation have very high marginal value and limited substitutes. This is reflected in ECHO-Global by the assumptions taken on the self-price elasticities of demand (-0.1 for the domestic sector and -0.54 for the industrial sector), indicating that increases in water prices would lead to higher reductions in water demand in the industrial sector than in the domestic sector. Therefore, it is not counter-intuitive that total economic value of water (net surplus) in the industrial sector is lower than that in the urban sector. It is, however, true that the economic water productivity of water in the industrial sector (measured by dividing the value added by the total water use) is often very high, but it doesn't reflect the economic value of water.

18. P33, L621: The manuscript states that a combination of water management options can help satisfy water demand. However, under the ENV scenario, both environmental flow requirements and constraints on non-renewable groundwater use are expected to reduce the water available for irrigation. Probably, this leads to substantial decreases in agricultural water withdrawals compared to BAU. It would be helpful for the authors to clarify whether this description is correct and whether these reductions result from unmet water demand due to supply constraints, or from economically optimal decisions under restricted water availability.

The reductions in agricultural water withdrawals under the ENV scenario are primarily driven by constraints established to maintain minimum environmental flows and minimize the use of non-renewable groundwater. Once these constraints are in place, we apply demand management options as described in the DM scenario, including improvements in irrigation efficiency and the economically optimal allocation of water among sectors and irrigated land among crops. These measures further reduce water consumption by enhancing efficiency and prioritizing high-value uses. We have clarified this distinction in Section 5.2, Lines 706-710 of the revised manuscript.

19. P34, L657: While the model provides a detailed representation of water allocation across BCUs within river basins, it does not appear to incorporate institutional or policy-based constraints such as transboundary water treaties or cooperative water management. Similarly, international trade in agricultural products is not modeled, despite its potential impact on regional cropping patterns and water demand. Clarifying these limitations would help define the scope and appropriate applications of the modeling framework.

ECHO-Global version 1.0 doesn't represent institutional or policy-based constraints. These constraints can be easily added to the model when data and quantitative information on those constraints would be available. To our knowledge, only qualitative information is available on international treaties.

Regarding international trade, ECHO-Global version 1.0 only represents current irrigated crop area and not rainfed crop area. It is also not able to project future changes in irrigated crop area, but rather simulate the effect of changes in water availability or other constraints and incentives (e.g., changing water prices, providing subsidies for investing in efficient irrigation systems) on current irrigated crop mix. In future work, the ECHO-Global can be further enhanced to consider

land use change and food trade, either through coupling with other sectoral models or incorporating additional constraints to e.g., enable the expansion of irrigated crop production.

We have now added additional text in section 5.3 on limitations of current ECHO-Global model version 1.0 and potential future extensions (lines 760-772).

**Minor comments**

20. P2 L41-46: Would you elaborate or rephase "appropriate water management options ... consistent across spatial scales"?

We meant that the wider impacts of the water management options beyond their implementation area need to be considered, because of the potential spillover effects driven by global processes such as international trade.

21. General: I suggest reorganizing Sections 2.2–2.4 so that the spatial delineation and data sources (currently in Sections 2.3 and 2.4) are presented before the model formulation (Section 2.2). This would help readers understand the origins and meaning of key parameters or assumptions—such as willingness to pay and irrigation efficiency—before encountering them in the description on equations. Presenting the data context first would improve the overall readability of the modeling framework.

Thank you for your suggestion. We believe that presenting the core model mathematical formulation first is more appropriate. However, we now tried to further explain parameters and variable used in the model. In addition, as suggested by reviewer #2 we have created a new table (Table A1) in the annex that presents all model sets, parameters and variables. We hope this is acceptable to this reviewer.

22. P.12, Eq. 25: the net present value (NPV) is defined as a summation over time ttt, yet the notation "Max NPV" is somewhat ambiguous. It might be clearer to explicitly show the double summation over both ttt and uuu, and to define NPV as a function of ttt, to clarify that it accumulates time-discounted net benefits across periods.

Corrected.

23. Figure 3: The colorbar needs unit.

Unit added in Figure 3.

24. Figure 5(a): Groundwater is shown as a single aggregated category. However, since the model differentiates between renewable and non-renewable groundwater—both conceptually and in terms of cost and sustainability—it would be more informative to distinguish these sources in the figure. This would also better support the interpretation of the ENV scenario, which specifically aims to reduce non-renewable groundwater use. I suggest disaggregating groundwater into renewable and non-renewable components to enhance the clarity and policy relevance of the figure.

In Figure 5a, groundwater sources are disaggregated into renewable and non-renewable components.

25. Figure 6: Maps appear to be vertically compressed, which may hinder the geographic interpretation of spatial patterns. The aspect ratio does not reflect the natural proportions of the Earth's latitude—longitude grid, making it difficult to compare regions and assess spatial trends accurately. I recommend adjusting the map projection or aspect ratio to improve visual clarity and ensure accurate geographic representation.

We have improved the visualization of maps in Figure 6.

---

## Referee Report (RR1)

**Reviewer #3 - General Assessment of the Revised Manuscript**

I would like to sincerely thank the authors for their thoughtful and comprehensive responses to my previous comments. It is clear that great effort has been made to carefully address nearly all of the points raised, both in the revised manuscript and in the accompanying point-by-point response. I greatly appreciate the authors' diligence in revising the text, figures, and explanations related to the model structure, parameters, and scenario assumptions.

The revised version of the manuscript reads much more clearly and cohesively, and I believe it is now in very good shape for publication as a model description paper in *GMD*. The improvements have significantly enhanced the transparency of the study, especially for readers who may be unfamiliar with certain modeling or economic concepts. I commend the authors for their extensive revisions and for their commitment to scientific clarity.

That said, I would like to respectfully request that the authors revisit a few remaining (previously major but now minor) issues related to my previous comments. These are relatively minor points, and I leave it to the authors' discretion to determine whether and how to address them, depending on necessity or feasibility. However, I believe that addressing them, where possible, would further improve the manuscript's precision and consistency.

**Remaining MINOR Points for Consideration:**

(Noe: Here, comment numbers are the same as ones in the previous review round)

- Major Comment 2: While the revised manuscript now includes an explanation of
  how initial reservoir storage levels are given, it still lacks any reference to dead
  storage in the text. If dead storage is assumed to be zero in the current version of
  the model, this assumption should be also clearly stated in the text.
- Major Comment 4: The addition of the discussion regarding the Ricardian rent principle is appreciated. However, it would be preferable to cite a relevant reference or source that supports this economic assumption, to enhance transparency for readers unfamiliar with the concept.
- Major Comment 18: The revision helpfully clarifies which components are
  included in the phrase "a combination of water management options." However,
  specifically, "limiting the use of non-renewable groundwater can help satisfy the

demand" still feels somewhat counterintuitive to me. If the intention is to suggest that such a constraint indirectly enables demand satisfaction by encouraging the adoption of other demand management options (as described in the DM scenario), then this causality could be clarified. Otherwise, limiting a water supply source seems to act more as just a constraint (as the authors also mention in the text) rather than as a facilitator in satisfying demand. In the authors' own explanation, it is the demand management options that directly serve to satisfy the water demand.

To avoid misunderstanding, especially in a scientific context, I would suggest rephrasing the sentence along the following lines. I hope the authors will consider revisiting this sentence to ensure the intended message is as precise and interpretable as possible.

"A combination of water management options (including improving irrigation efficiency and optimizing land and water demand allocation, even with limiting use of non-renewable groundwater) can reduce the water demand to help satisfy the demand."

---

## Author Response (AR2)

**Response letter to review comments of the manuscript** "Development of the global hydroeconomic model (ECHO-Global version 1.0) for assessing the performance of water management options"

**Editor:**

Dear Dr. Kahil and co-authors,

Thanks for preparing such a thorough revision of your manuscript. I have received reports from all three reviewers. Please consider the minor revisions requested by Reviewer 3. Reviewers 1 and 2 were satisfied with the edits you already provided.

Thanks,

Tom

Dear Editor,

Foremost, we would like to thank you and the reviewers for handling our manuscript and providing thoughtful comments throughout the review process that have helped us to improve substantially the quality of the paper. We have now responded to the comments of the referee #3. The revisions we have made are highlighted in the text and our responses to the referee's comments can be found below.

Best Regards,

Taher Kahil on behalf of authors

**Referee #3:**

**General Assessment of the Revised Manuscript:**

I would like to sincerely thank the authors for their thoughtful and comprehensive responses to my previous comments. It is clear that great effort has been made to carefully address nearly all of the points raised, both in the revised manuscript and in the accompanying point-by-point response. I greatly appreciate the authors' diligence in revising the text, figures, and explanations related to the model structure, parameters, and scenario assumptions.

The revised version of the manuscript reads much more clearly and cohesively, and I believe it is now in very good shape for publication as a model description paper in GMD. The improvements have significantly enhanced the transparency of the study, especially for readers who may be unfamiliar with certain modeling or economic concepts. I commend the authors for their extensive revisions and for their commitment to scientific clarity.

That said, I would like to respectfully request that the authors revisit a few remaining (previously major but now minor) issues related to my previous comments. These are relatively minor points, and I leave it to the authors' discretion to determine whether and how to address them, depending on necessity or feasibility. However, I believe that addressing them, where possible, would further improve the manuscript's precision and consistency.

We are pleased to know that our revision was acceptable to you. Thank you for your thoughtful additional comments. We have now addressed them. The revisions we have made are highlighted in the text and our responses to the comments can be found below.

1. Major Comment 2: While the revised manuscript now includes an explanation of how initial reservoir storage levels are given, it still lacks any reference to dead storage in the text. If dead storage is assumed to be zero in the current version of the model, this assumption should be also clearly stated in the text.

Thank you for your comment. We have now indicated in the revised manuscript that dead storage is assumed to be zero, in the absence, to our knowledge, of available global data or references on this parameter. We found only one study of Zhao et al. (GMD, 2024), included now in the reference list, which uses the same assumption of minimum reservoir storage equal to zero. Nevertheless, this parameter can be easily adjusted in ECHO-Global, especially when the model is applied to specific basins where data on reservoir features and operations might be available.

2. Major Comment 4: The addition of the discussion regarding the Ricardian rent principle is appreciated. However, it would be preferable to cite a relevant reference or source that supports this economic assumption, to enhance transparency for readers unfamiliar with the concept.

In the ECHO-Global model, we use a variant of the PMP procedure developed by Dagnino and Ward (2012) to calibrate the model to replicate observed land and water allocations in the reference year. This PMP variant assumes that crop yield is a decreasing function of the amount of land in production. This assumption is consistent with the Ricardian theory of rent indicating how the price of land is determined based on its fertility and location, leading to differences in rent across different parcels of land. This theory has been developed in 1817 by economist David Ricardo. In our paper, we would like to refer readers to the study of Dagnino and Ward (2012) that developed the variant PMP procedure and provided a clear demonstration of it.

3. Major Comment 18: The revision helpfully clarifies which components are included in the phrase "a combination of water management options." However, specifically, "limiting the use of non-renewable groundwater can help satisfy the demand" still feels somewhat counterintuitive to me. If the intention is to suggest that such a constraint indirectly enables demand satisfaction by encouraging the adoption of other demand management options (as described in the DM scenario), then this causality could be clarified. Otherwise, limiting a water supply source seems to act more as just a constraint (as the authors also mention in the text) rather than as a facilitator in satisfying demand. In the authors' own explanation, it is the demand management options that directly serve to satisfy the water demand.

To avoid misunderstanding, especially in a scientific context, I would suggest rephrasing the sentence along the following lines. I hope the authors will consider revisiting this sentence to ensure the intended message is as precise and interpretable as possible.

"A combination of water management options (including improving irrigation efficiency and optimizing land and water demand allocation, even with limiting use of non-renewable groundwater) can reduce the water demand to help satisfy the demand."

Thank you for this suggestion. We agree with it and we have rephrased the indicated sentence as suggested.